# A scenario for an evolutionary selection of ageing

**Tristan Roget[1], Claire Macmurray[2], Pierre Jolivet[3], Sylvie Meleard[4], Michael Rera[5]\***

[1]Institut Montpelliérain Alexander Grothendieck (IMAG), Université de Montpellier, Montpellier, France; [2]Université Paris Cité, Paris, France; [3]Sorbonne Université, CNRS, Paris, France; [4]Institut Universitaire de France et École Polytechnique, CNRS, Institut polytechnique de Paris, Palaiseau, France; [5]Université Paris Cité, Institut Jacques Monod, Paris, France

## eLife assessment

Through a theoretical approach, this study makes **important** contributions to our understanding of the evolutionary causes of the ageing process. Using a simple individual-based model and computational simulations, the authors provide **convincing** evidence that ageing can be a trait under natural selection, opening the door for further discussion in the context of lifespan extension research.

**\*For correspondence:**
michael.rera@cnrs.fr

**Competing interest:** The authors declare that no competing interests exist.

**Abstract** Signs of ageing become apparent only late in life, after organismal development is finalized. Ageing, most notably, decreases an individual's fitness. As such, it is most commonly perceived as a non-adaptive force of evolution and considered a by-product of natural selection. Building upon the evolutionarily conserved age-related Smurf phenotype, we propose a simple mathematical life-history trait model in which an organism is characterized by two core abilities: reproduction and homeostasis. Through the simulation of this model, we observe (1) the convergence of fertility's end with the onset of senescence, (2) the relative success of ageing populations, as compared to non-ageing populations, and (3) the enhanced evolvability (i.e. the generation of genetic variability) of ageing populations. In addition, we formally demonstrate the mathematical convergence observed in (1). We thus theorize that mechanisms that link the timing of fertility and ageing have been selected and fixed over evolutionary history, which, in turn, explains why ageing populations are more evolvable and therefore more successful. Broadly speaking, our work suggests that ageing is an adaptive force of evolution.

## Introduction

Ageing is the umbrella term used to describe the processes that take place when an organism's capacity to thrive diminishes with time. Patterns of ageing vary greatly given the organism, from negligible senescence to post-reproductive death through progressive age-dependent mortality increase (*Jones et al., 2014*). While ageing, as an observable process, is evident, the evolutionary role of ageing is unclear and conceptually challenging. An ageing individual is less fit, nevertheless, ageing seems to be broadly present through evolutionary time. Our work aims to explore the question, "is this mere chance– is ageing strictly a by-product of other things under selection– or is it somehow adaptive"? Soon after Charles Darwin published his theory of evolution, August Weismann situated ageing within this framework (*Weismann, 1882*) by theorizing that, "there exists a specific death-mechanism designed by natural selection to eliminate the old, and therefore worn-out members of a population" (*Gavrilov and Gavrilova, 2002*). Since then, however, it is mostly accepted that "ageing

**eLife digest** It is a question as old as Darwin's theory of evolution itself: how is ageing affected by natural selection? The prevailing view is that the process of biological ageing is not adaptive and therefore not directly subject to selection pressures. Take for example a gene causing a fatal disease late after an average individual had reproduced, thus being passed on to the next generation despite its detriment to the individual. This suggests that natural selection acts less strongly on such genes, which can therefore accumulate and cause aging if they do not impact an organism's reproductive fitness earlier in life.

However, many studies have shown that specific genes control an animal's lifespan and the onset of ageing through evolutionarily conserved mechanisms. For example, in fruit flies, aging can be categorised into two distinct phases determined by the manifestation of the so-called Smurf phenotype associated with accelerated signs of ageing and an increased risk of death. A pattern where the offspring of older parents live less long than those of younger parents has also been observed across species, also known as the Lansing effect. In this case, ageing can affect the reproductive success of future generations and can therefore be subject to selection pressures.

Roget et al. looked at the trade-offs between an individual's reproduction and homeostasis using a mathematical model to address whether the distinct phases of aging – as seen in the Smurf phenotype – can appear and be maintained throughout evolution. Using a mathematical model, Roget et al. simulated individuals possessing only one copy of two genes. One controls the duration of reproductive ability, and the other defines the age at which the risk of death becomes non-zero.

This revealed that a simple hypothetical haploid and asexually reproducing system can evolve a life history separated into two phases in the computer simulations. Interestingly, the modelled organisms evolved in a way that the duration of reproduction exceeded the homeostatic maintenance duration. This generated a phase where individuals are capable of reproduction with a high risk of death, similar to the previously described Smurf phase. Roget et al. observed that aging populations showed a lower risk of extinction than non-aging ones, as well as an increased genetic variability of the offspring.

The apparent benefits of ageing in this model imply that ageing can be an adaptive force of evolution and subject to positive selection or, at least less negative selection than expected. This minimal model helps explain trade-offs between reproduction and homeostatic maintenance during evolution. Further work may include parameters such as sexual reproduction and multiple gene copies.

is not adaptive since it reduces reproductive potential" (*Kirkwood and Holliday, 1979*) and hence, fitness. Weismann's own theories eventually evolved to more closely represent this current position.

At present, ageing is typically viewed as a 'side-effect', or byproduct, of other processes under selection (*Fabian, 2011*), which implies that ageing, or the mechanisms that cause ageing, are neither selected nor adaptive— precisely as capacities that would prove advantageous for a given population. This view took precedent starting in the 1950s and it is now assumed that the genetics or molecular processes that drive ageing help to explain how ageing has evolved (*Gavrilov and Gavrilova, 2002*). Peter Medawar's theory of *mutation accumulation* defends that ageing is caused by the progressive accumulation of deleterious mutations with effects that show only late in life (*Medawar, 1953*). Williams' *antagonistic pleiotropy* theory goes further than Medawar's by presupposing the existence of antagonistic genes and mutations: beneficial at an early age, these genes/mutations prove disadvantageous at a later age (*Williams, 1957*). Evolutionary conserved genes involved in both the regulation of longevity and organismal growth were discovered in the model organism *Caenorhabditis elegans* (*Kenyon et al., 1993*) and later shown to be conserved in flies (*Clancy et al., 2001*), mice (*Blüher et al., 2003*), and humans (*van Heemst et al., 2005*). Thus, genetic modulators for longevity exist and express themselves through evolutionarily conserved physiological mechanisms. With genes involved in the onset of longevity, there is a potential substrate for selective pressure to apply. Regardless, it is generally accepted that ageing is *neither* a programmed nor beneficial trait for species (*Kowald and Kirkwood, 2016*).

The Smurf phenotype is a simple age-associated intestinal permeability phenotype that was first observed in *Drosophila* (*Rera et al., 2011*). Evolutionarily conserved in nematodes, zebrafish

(*Dambroise et al., 2016*). and mice (*Cansell et al., 2023*), this phenotype allows for the identification of two distinct subpopulations– non-Smurf individuals and Smurf ones– at any time in a given population. All individuals undergo the transition (from non-Smurf to Smurf) prior to death (*Rera et al., 2012*; *Tricoire and Rera, 2015*). In flies, the Smurf phase is characterized by multiple physiological marks of ageing such as the high risk of impending death, loss of energy stores, systemic inflammation, reduced motility (*Rera et al., 2012*), and reduced fertility (*Rera et al., 2018*). More generally, the transcriptional hallmarks (*Frenk and Houseley, 2018*) usually associated with ageing are mostly observed in the latter phase (*Zane et al., 2023*). To summarize, this phenotype allows for the identification of two successive and necessary phases of life with all the age-related changes occurring in the last. Motivated by these biological observations, we recently assessed (*Méléard et al., 2019*) the possibility of obtaining, over time, such two phases of life. In order to simplify, we decided to consider the evolution of such a process in a bacteria-like organism, through the design and implementation of an asexual and haploid age-structured population mathematical model. We constrained the evolutive trajectory of ageing (within this model) through the Lansing effect– a transgenerational effect impacting longevity. Smurf individuals carry the propensity to demonstrate this effect. The Lansing effect is a transgenerational phenomenon, first described by Albert Lansing in the late 1940s, whereby the 'progeny of old parents do not live as long as those of young parents' (*Lansing, 1954*; *Lansing, 1947*). This was first observed in rotifers. More recently, it has been shown that older *Drosophila* females, and to some extent males tend to produce shorter lived offspring (*Priest et al., 2002*). Older zebra finch males give birth to offspring with shorter telomere lengths and reduced lifespans (*Noguera et al., 2018*). In humans, 'older father's children have lower evolutionary fitness across four centuries and in four populations' (*Arslan et al., 2017*). Despite the absence of consensus regarding any underlying mechanism, the Lansing effect is broadly conserved and therefore relevant (*Monaghan et al., 2020*). We observed, through this Lansing-positive model, that the ageing phase overlaps with the pre-ageing phase in evolutionary time.

Here, we decided to generalize this model to any system able to reproduce and maintain homeostasis, without the necessary constraint of the Lansing effect, and in hopes of understanding how such a two-phase ageing process might have evolved. We thus show the following:

1. Through time, the end of the healthy phase and the beginning of the senescent phase converge even in the absence of a transgenerational effect (the Lansing effect).
2. With an equal Malthusian parameter at $t_0$, Lansing populations are more successful than non-Lansing populations, suggesting that the individual loss of fitness is compensated at the population level.
3. Ageing (or senescence-carrying) populations are more evolvable than non-ageing populations. We theorize this is because ageing populations are quicker to explore genotypic space.

This is all to suggest that ageing is, as a function, decreasing both reproductive and homeostatic capabilities of an organism, both an attractor configuration and an adaptive force of evolution, in opposition to what is most commonly assumed.

## Results

The model (called bd model) and its population dynamics follows those described in *Méléard et al., 2019*. Briefly, the model delineates an asexual and haploid population, structured by a life-history trait that is defined by a pair of parameters - genes - $(x_b, x_d)$ where $x_b$ defines the fertility span and $x_d$, the age at which the mortality risk becomes non-null. Here, we generalized the model to any intensities of birth and death denoted $(i_b, i_d)$ as well as to populations without Lansing effect (*Figure 1*, see also Appendix 1). The selective pressure is enforced by a logistic competition $c$ mimicking a maximum carrying capacity of the environment, thus no explicit adaptive value is given to any particular trait. Additionally, for each reproduction event, a mutation ($h$) of probability $p$ can affect both genes $x_b$ and $x_d$ independently, following a Gaussian distribution centered on the parental trait. In *Figure 1*, the different cases are explored, depending on the respective values of $x_b$ and $x_d$. Individuals in the *Figure 1b–c* configuration (for $x_b \leq x_d$) will always give progeny with a genotype $(x_b, x_d) \mp (h_b$ and/or $h_d)$. The evolutionary outcome of individuals carrying a genotype with $x_d < x_b$ (*Figure 1a*) is slightly more nuanced and depends on the parental age $a$ *and* whether the parent carries the possibility for a Lansing effect or not (*Figure 1d–f*). If $a < x_d$, or if the parent does not carry a Lansing effect, the

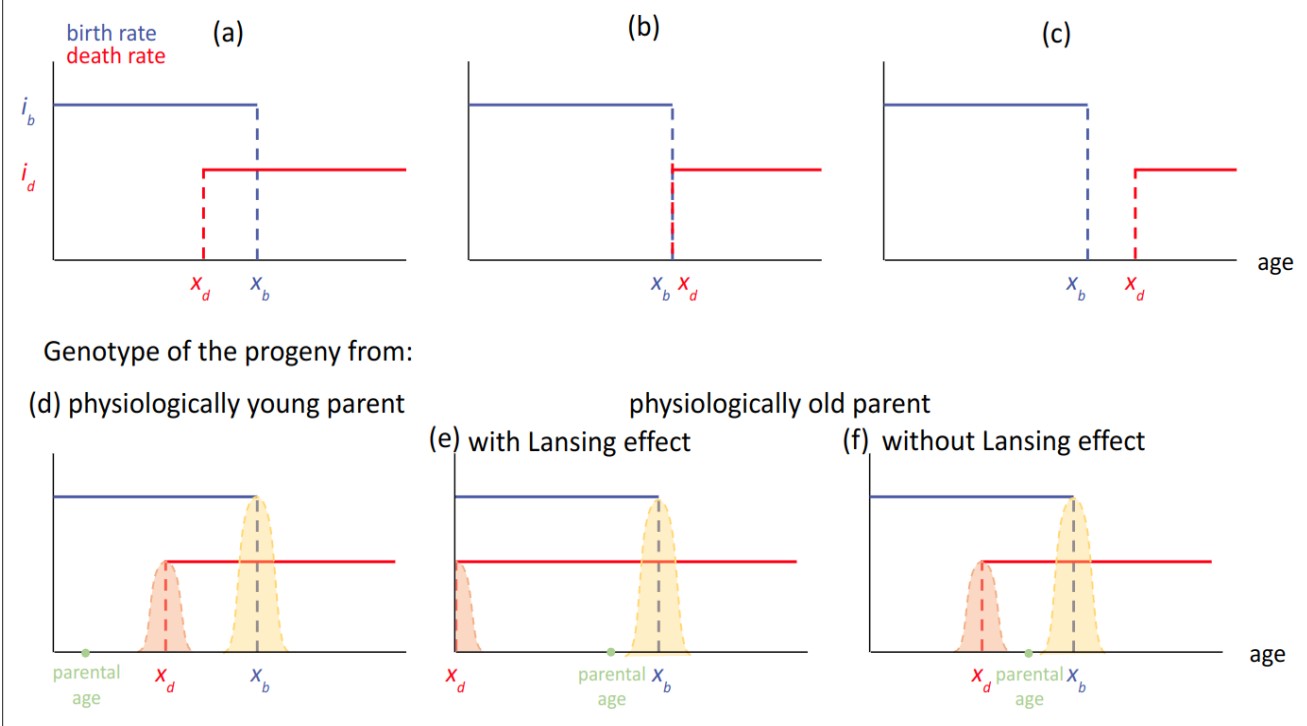

**Figure 1.** Three typical configurations of the model with $i_b > i_d$ and their effect on progeny's genotypes as a function of parental age. (upper panel) Each haploid individual is defined by a parameter $x_b$ defining its fertility span of intensity $i_b$ and a parameter $x_d$ defining the time during which it will maintain itself, with an intensity $i_d$. These parameters can be positive or null. (a) 'Too young to die': it corresponds to configurations satisfying $x_d < x_b$. (b) 'Now useless': it corresponds to configurations where $x_b = x_d$. (c) 'Menopause': it corresponds to configurations where $x_d > x_b$. (lower panel) Each individual may randomly produce a progeny during its fertility span [0; $x_b$]. (d) In the case of physiologically young parents ($a < x_d$), the progeny's genotype is that of its parent ∓ a Gaussian kernel of mutation centered on the parental gene. In the case of the reproduction event occurring after $x_d$, for configuration (a) above, two cases are observed, (e) if the organism carries a Lansing effect ability, the $x_d$ of its progeny will be strongly decreased. (f) In the absence of the Lansing effect, the default rule applies.

genotype of the progeny will be as previously described. But if $a > x_d$, and if the parent carries the Lansing effect, the progeny then inherits a dramatically reduced $x_d$ (here $x_d$ is set to 0), mimicking a strong Lansing effect.

In our previous work (**Méléard et al., 2019**), we formally and numerically showed the long-time evolution of the model to converge towards ($x_b - x_d$)=0 in the case of individuals carrying a Lansing effect. Here, we explore the convergence of ($x_b - x_d$) without the strong transgenerational effect of ageing. We implemented a new version of the model, devoid of the Lansing effect, and simulated its evolution for a viable - that is allowing the production of at least one progeny - trait ($x_b = 1.2$, $x_d = 1.6$). Surprisingly, we still observe a convergence of ($x_b - x_d$) in finite time. The dynamics of the trait ($x_b$, $x_d$) is described by the canonical equation of adaptive dynamics, which depends on the Malthusian parameter and its gradient (Appendix 1). The Malthusian parameter can be interpreted as the age-specific strength of selection (**Hamilton, 1966**). The speed at which $x_b$ and $x_d$ evolves, decreases with time, just as in the previous form of the model (**Méléard et al., 2019**), allowing us to recover the well-observed, age-related decrease in the strength of selection (**Haldane, 1941**; **Hamilton, 1966**; **Medawar, 1953**). Simulations of the generalized bd model presented here show that the $x_b - x_d$ distance (the time separating the end of fertility from the increasing risk of death) converges, for any initial trait, towards a positive constant. Thus, the long-term evolution of such a system is a configuration similar to **Figure 2a** ($x_d < x_b$). The formal analysis of the generalized bd model confirms that the long-time limit of the traits ($x_b - x_d$) is the positive constant (defined by the formula in **Figure 2b**, mathematical analysis presented in Appendix 1), reached after a few dozen simulated generations (**Figure 2c**). Although we formally demonstrate the long-time limit for any $i_b$ and $i_d$, all our simulations are run using $i_b = i_d = 1$, in order to limit the number of conditions to assess and report. Surprisingly, the limit value of the trait is not affected by $x_b$ or $x_d$ values - the fertility span and mortality per se - but only

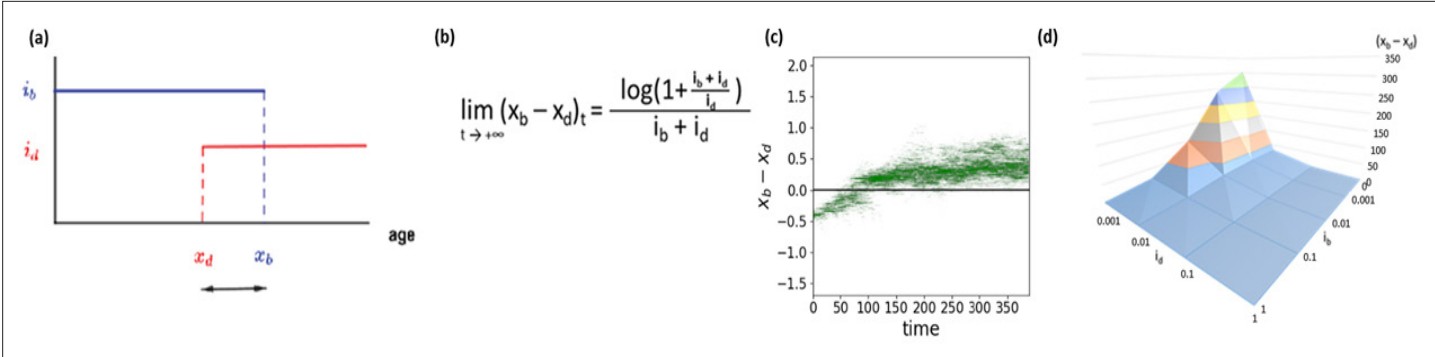

**Figure 2.** The bd model shows a convergence of $x_b$ - $x_d$ towards a positive value. Dynamics of the individual-based model shows a convergence of $x_b$ - $x_d$ towards a positive constant value in the absence of the Lansing effect. (**a**) The generalized b-d model shows a convergence of $(x_b - x_d)$ for any $i_b$ and $i_d$ towards a positive value given by (**b**) (Annexe 4.3, **Figure 2**). (**c**) Simulation of 1000 individuals with initial trait ($x_b$=1.2, $x_d$=1.6) of intensities $i_b$=$i_d$=1, a competition c=0.0009 and a mutation kernel ($P$=0.1, $\sigma$=0.05) show that the two parameters co-evolvetowards $x_b$ - $x_d \cong 0.55$ that is log(3)/2. (**d**) Landscape of solutions ($x_b$ - $x_d$) as a function of $i_b$ and $i_d$ (colors separate ranges of 50 units on the z-axis).

by their respective intensities $i_b$ and $i_d$. These intensities can be interpreted as the instant mortality risk $i_d$ and the probability to give a progeny $i_b$. Interestingly, the long-time limit values for any $i_b$ and $i_d$ shows a significantly stronger sensitivity to the increasing mortality risk $i_d$ than to reproduction by almost two orders of magnitude (**Figure 2d**). In addition, for extremely low values of $i_b$ and $i_d$ - that is below 0.01 - the apparent time correlation of the fertility span and mortality onset is almost nonexistent; this is because ($x_b$ - $x_d$) is large. Biologically, this would appear to an observer as the mortality onset occurring long before the exhaustion of reproductive capacity. Such an organism would be thus characterized as having no significant fertility decrease during the ageing process. On the other end, for individuals showing either a high instant mortality risk or a high probability to give a progeny, the ($x_b$ - $x_d$) trait is close to 0, meaning that fertility and organismal integrity maintenance are visibly - that is observable by an experimenter - correlated. It is important to note that this mathematical study concerns individuals for which the mean number of descendants per individual is large enough, allowing us to define a viability set of traits ($x_b$, $x_d$) (see data availability statement). Because of these mathematical properties, a tradeoff emerges between $i_b$, $i_d$, $x_b$ and $x_d$. Let's consider an organism - for both the Lansing and non-Lansing cases - with a low reproductive intensity $i_b$=0.01 and $i_d$=1. For this organism to propagate, the product $i_b * x_b$ has to be strictly superior to 1, hence here $x_b \geqslant 100$ (see data availability statement). In this example, the long-time limit of the trait ($x_b$ - $x_d$) is equal to log(2), thus $x_b$ and $x_d$ are of the same order of magnitude. With the same reasoning, the long-time evolution lower limit of ($x_b$ - $x_d$), of an organism that is significantly more fertile (with $i_b$=1, $i_d$=1), is $1/\sqrt{3}$. This model thus allows an elegant explanation for the apparent negative correlation previously described between longevity and fertility without the need of implementing energy trade-offs or relative efficiency of energy allocation between maintenance and reproduction (see data availability statement - examples).

In our model, regardless of the initial trait ($x_b$, $x_d$) in the viability set, evolution leads to a configuration of the trait such that the risk of mortality starts to increase before the fertility span ends. Similar to biochemical reactions involved in a given pathway that are evolutionarily optimized (e.g. through tunneled reactions and gated electron transfers), we hypothesize here that such a configuration, caused by simple mathematical constraints, creates the conditions for the apparition, selection, and maintenance of a molecular mechanism coupling $x_b$ and $x_d$. Such a coupling mechanism could thus be the so-called Lansing effect— the only described age-related decline in progeny's fitness that seems to affect numerous iteroparous species (**Lansing, 1947**; **Monaghan et al., 2020**). We assessed the likelihood of survival of an organism carrying such a non-genetic and pro-senescence mechanism when in competition with a population devoid of such a mechanism. To do so, we examined a population divided into two sub-populations: one made of individuals subject to the Lansing effect and the other made up of individuals not subject to the effect. We assume, as before, that each individual is under the same competitive pressure. The two initial sub-populations have the same Darwinian fitness approximated by their Malthusian parameter (see data availability statement, **Figure 3—figure**

**Table 1.** Populations with Lansing effect are favorably selected under logistic competition when the mutation rate is non-zero.
p is the mutation rate and c the intensity of the logistic competition. For each couple (p, c), 100 independent simulations were run with 500 individuals per population at t0 of which traits are (1.5; 1.3) Lansing and (1.5; 0.83)non-Lansing so that their respective Malthusian parameters are equal. Each simulation corresponds to 2.105 events of birth or death. Table (a) shows the ratio of Lansing and non-Lansing populations (out of 100 simulations in each case) that did collapse by the end of the simulation. For the lowest competition, none of the populations collapsed within the timeframe of simulations (-). For an intermediate value of competition, approximately less than half of Lansing populations disappear, relative to non-Lansing ones. Table (b) shows the ratio of the number of individuals generated between Lansing and non-Lansing populations. On average, Lansing populations generate approximately twice as many individuals as non-Lansing ones. (c) On average, Lansing populations grow 20% more than the non-Lansing. Values highlighted in green are discussed further below.

| | | **Mutation probability** | | | | | | | | | | | |
| | | 0 | 0.1 | 0.5 | 1 | 0 | 0.1 | 0.5 | 1 | 0 | 0.1 | 0.5 | 1 |
| | | **a) Lansing/non-Lansing collapsed population** | | | | **b) Lansing/non-Lansing number of individuals** | | | | **c) Lansing/total population size** | | | |
| Competition | $9.10^{-5}$ | - | - | - | - | 1.30 | 1.38 | 1.39 | 1.35 | 0.57 | 0.64 | 0.62 | 0.59 |
| | $9.10^{-4}$ | 1.02 | 0.62 | 0.56 | 0.66 | 1.70 | 2.84 | 3.58 | 3.20 | 0.49 | 0.62 | 0.60 | 0.55 |
| | $9.10^{-3}$ | 1.00 | 1.05 | 1.13 | 1.03 | 1.05 | 1.31 | 1.56 | 1.84 | - | 0.43 | 0.44 | 0.49 |

The online version of this article includes the following source data for table 1:

**Source data 1.** The magnitude of the Lansing effect does not influence the outcome of evolution.

supplement 1). Their traits are thus (1.5; 1.3)$_{Lansing}$ and (1.5; 0.83)$_{non-Lansing}$. In order to simplify the analysis, both the birth and death intensities are as follows: $i_b=i_d=1$ (the model is nevertheless generalized to any ($i_b$; $i_d$), see data availability statement). We simulated the evolution of such mixed populations for discrete pairs of mutation rate (p) and competition (c) parameters. Three indexes were calculated for each set of simulation: (*Table 1*) the ratio of Lansing and non-Lansing populations that collapsed ("-" indicates that all survived), (*Table 1*) the ratio of total number of progenies produced during the simulation by each population and (*Table 1*) the relative proportion of the Lansing population at the end of the simulation (*Table 1—source data 1*). Our 1200 simulations, each with $2.10^5$ birth-death events, summarized in *Table 1*, show that the Lansing populations survive at least as well as non-Lansing ones (*Table 1*) especially for a moderate competition parameter (c=$9.10^{-4}$) and low (in our simulations) mutation rate (*P*=0.1). With such conditions, Lansing populations show almost half the risk of disappearance of non-Lansing ones (*Table 1*), producing nearly three times as many descendants as non-Lansing populations (*Table 1*), for up to a 20% faster growing population (*Table 1*). The plots for a single. Thus, although the Lansing effect gives way to a significant proportion of progeny with an extremely low fitness ($x_d = 0$), pro-ageing populations show a decrease in the risk of collapse. Moreover, we observe a slightly better growth of the population, independent of the magnitude of the Lansing effect (*Figure 3—figure supplement 2*).

In order to understand the evolutionary success of a characteristic that seems to decrease an organism's fitness, we computed the average Malthusian parameter of each population through time. We had previously identified that this intermediate set of c and p was associated with the highest success rate of Lansing bearing populations and presented the results for this set (highlighted in green, *Table 1*). First, we observe that, on average, Lansing populations (blue) grow while non-Lansing ones (red) decrease in size (*Figure 3a* - blue and red curves represent deciles 1, 5, and 9). In the simulations where both populations coexist, the higher fitness of the Lansing population is marginal, with these populations growing 20% more than the non-Lansing population (*Figure 3b*). This higher success rate seems to be driven by a faster and broader exploration of the Malthusian parameter space in the Lansing population (*Figure 3c*). This maximization of the Malthusian parameter is not associated with any significant difference of individual lifespan (time of death - time of birth) distributions of either population (*Figure 3d*). Although subjected to the same competition **c**, the distribution of the progeny from non-Lansing populations is essentially that of the parental trait in the first 5 generations, while Lansing progenies (not affected by the Lansing effect; we excluded progeny with $x_d=0$ for the comparison) explore a broader part of the trait space (*Figure 3e*). Interestingly, low fitness progeny ($x_d=0$) represents up to 10% of the population for a significant amount of time (*Figure 3f*).

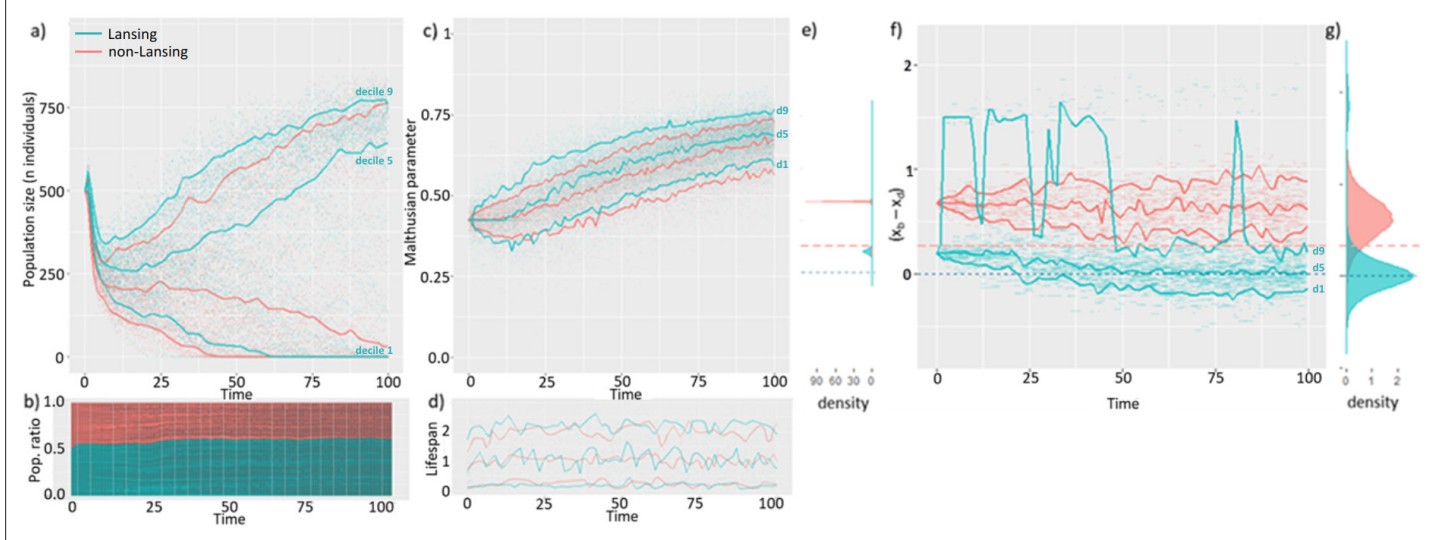

**Figure 3.** The Lansing effect maximizes populational survival by increasing its evolvability. 100 independent simulations were run with a competition intensity of $9.10^{-4}$ and a mutation rate p=0.1 on a mixed population made of 500 non-Lansing individuals and 500 individuals subjected to such effect. At $t_0$, the population size exceeds the maximum load of the medium thus leading to a population decline at start. At $t_0$, all individuals are of age 0. Here, we plotted a subset of the $100.10^6$ plus individuals generated during the simulations. Each individual is represented by a segment between its time of birth and its time of death. In each graph, blue and red curves represent deciles 1, 5, and 9 of the distribution at any time for each population type. (**a**) The higher success rate of Lansing bearing populations does not seem to be associated with a significantly faster population growth but with a lower risk of collapse. (**b**) For cohabitating populations, the Lansing bearing population (blue) is overgrowing by only 10% the non-Lansing one (red). (**c**) This higher success rate is associated with a faster and broader exploration of the Malthusian parameter - surrogate for fitness - space in Lansing bearing populations (**d**) that is not associated with significant changes in the lifespan distribution (**e**) but a faster increase in genotypic variability within the [0; 10] time interval. (**f**) This occurs although progeny from physiologically old parents can represent up to 10% of the Lansing bearing population and leads to it reaching the theoretical optimum within the timeframe of simulation (**g**) with the exception of Lansing progenies. (**e–g**) Horizontal lines represent the theoretical limits for ($x_b$ - $x_d$) in Lansing (blue) and non-Lansing (red) populations.

The online version of this article includes the following figure supplement(s) for figure 3:

**Figure supplement 1.** Evolution of the average Malthusian parameter value in Lansing and non-Lansing populations as a function of time.

**Figure supplement 2.** Evolution of the Lansing and non-Lansing populations size as a function of time.

As a consequence, Lansing populations reach the equilibrium trait faster than the non-Lansing ones (*Figure 3g*). Thus, the relatively higher success rate of Lansing bearing populations seems to be associated with a higher genotypic diversity. This, in theory, leads to a broader range of fitness types. The 'optimal' fitness is therefore achieved earlier (or more easily), thus explaining the relative success of the population. This is an example of a population that demonstrates a greater ability to evolve (i.e. the population 'possesses' the attribute termed 'evolvability').

Our model explains, in mathematical terms, why the mortality onset is evolutionarily linked to reproductive mechanisms (or fertility). Nevertheless, the numerical exploration of our model's behavior has been limited so far to initial conditions, where the competing populations were of equal Malthusian parameters. The low number of generations involved suggests that the conditions for the development, selection, and maintenance of mechanisms of ageing (*Lemoine, 2021*) occurs early on in evolutionary history, in a population of mixed individuals. As such, we decided to test the evolution of the trait ($x_b$ - $x_d$) in Lansing and non-Lansing bearing individuals of uniformly distributed traits on [−10;+10] (*Figure 4* - left panel). We chose to plot one (*Figure 4* - central panel) of the hundred simulations we made. This simulation is representative of the general results. Simulations show, in over 110 million individuals, an early counter-selection of extreme trait values, typically ($x_b$ - $x_d$)>4. Interestingly, the whole space of ($x_b$ - $x_d$) trait is not explored evenly and the positive part of the trait space represents approximately 2/3 of the individuals (although the branched evolution process led to both the positive ('Too young to die' – *Figure 1a*) and negative ('Menopause' – *Figure 1c*) sides of the trait space). Both the Lansing and non-Lansing bearing populations manage to co-exist until the end of the simulation, each reaching a distribution centered on their respective theoretical solutions (*Figure 4* - right panel): 0 for the Lansing (*Méléard et al., 2019*) and log(3)/2 for the non-Lansing. In this context, where the

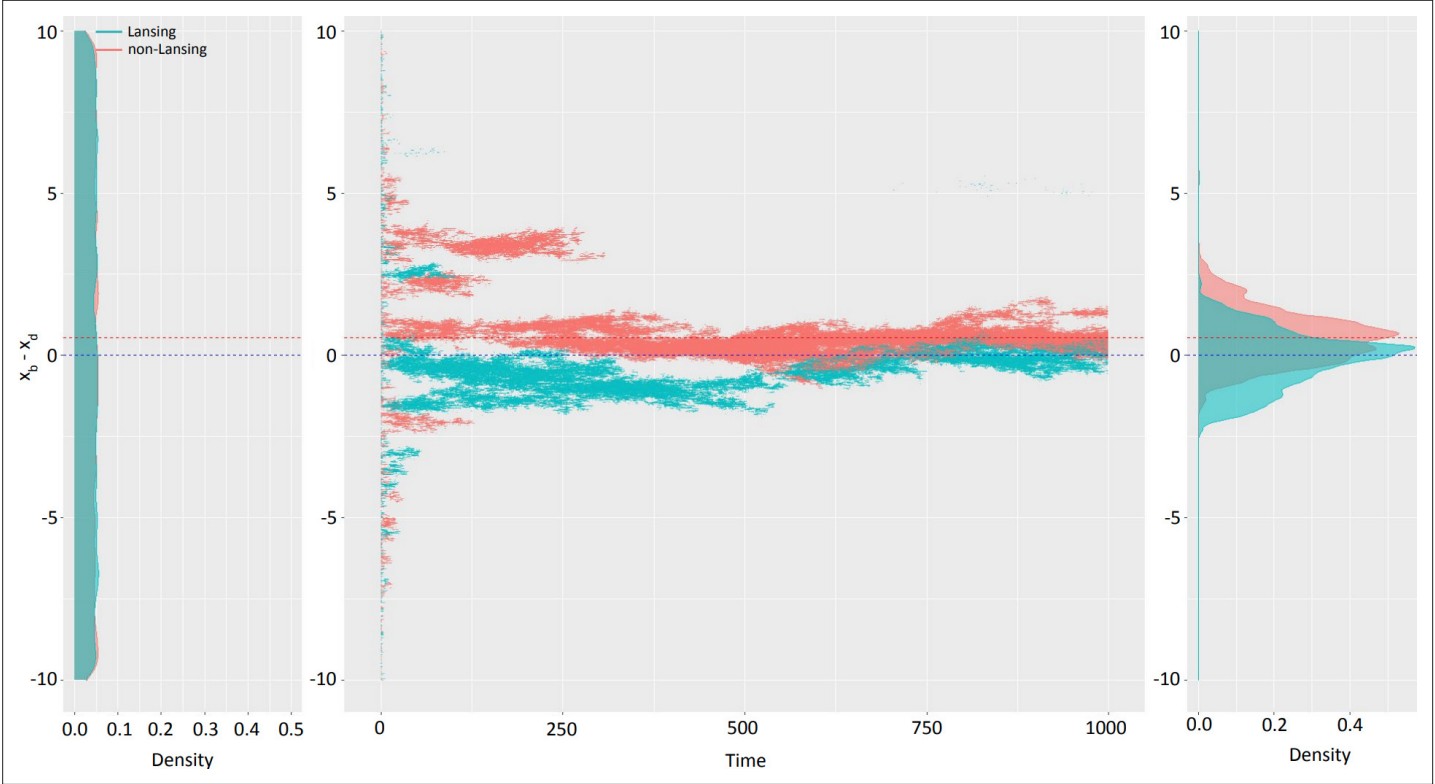

**Figure 4.** Mixed populations lead to ($x_b$ - $x_d$) theoretical limit in a limited time and cohabitation of Lansing and non-Lansing populations. Starting with a homogenous population of 5000 Lansing bearing and 5000 non-Lansing individuals with traits uniformly distributed from –10 to +10 (left panel), we ran 100 independent simulations on time in [0; 1000]. (center panel) Plotting the trait ($x_b$ - $x_d$) as a function of time for one simulation shows a rapid elimination of extreme traits and branching evolution. (right panel) The final distribution of traits in each population type is centered on the theoretical convergence limit for each. $N_{total} \cong 110$ million individuals, $c=9.10^{-4}$, $p=0.1$.

initial condition does not restrict the competition to individuals of identical Malthusian parameters, the Lansing bearing population is significantly less successful than the non-Lansing one (representing only one third of the final population size). As such, the evolution of a mixed population of individuals with a trait ($x_b$ - $x_d$) initially uniformly distributed on [–10;+10], with or without a strong intergenerational effect, will lead to a mixed solution of individuals carrying a trait that converges towards the theoretical solution (such as $x_d \lesssim x_b$), thus allowing the maximization of fertility without cluttering the environment with non-fertile individuals. This result is very similar to Weismann's first intuition (**Weismann, 1882**). Nevertheless, this interpretation seems somehow finalist (i.e. presumes that the effects necessitate the causes) and does not yet discriminate *why* the Lansing population is evolutionarily successful in comparison to the non-Lansing population. Thus, we next explore the parameter of evolvability further, which leads us to yet again conceptualize ageing as an adaptive trait.

Populations that consist of Individuals who can transmit ageing 'information' to the next generation are relatively more successful, within the framework of our model. Thus, to understand the origin of this pattern, we examined the differential landscape of the Malthusian parameters as a function of the trait ($x_b$, $x_d$) for both Lansing and non-Lansing populations. We built this landscape numerically using the Newton method (see data availability statement). First, it is interesting to notice that, from the equations, we have derived the maximum rate of increase for Malthusian parameters, this being $1/i_d$ with a maximum fitness value capped by $i_b$ (data availability statement). Consistent with our previous characterization of the Trait Substitution Sequence in populations with Lansing effect (**Méléard et al., 2019**), Lansing individuals have a symmetrical fitness landscape (**Figure 5**, blue lines) centered on the diagonal $x_b = x_d$ (**Figure 5**, green diagonal). Along the latter, we can directly observe what is responsible for a 'selection shadow'. As $x_b$ and $x_d$ increase, a mutation of the same magnitude has smaller and smaller effects on the fitness, thus allowing the accumulation of mutations (**Figure 5**, blue arrows). The graphical representation of non-Lansing individuals is asymmetric— the rupture of symmetry occurs

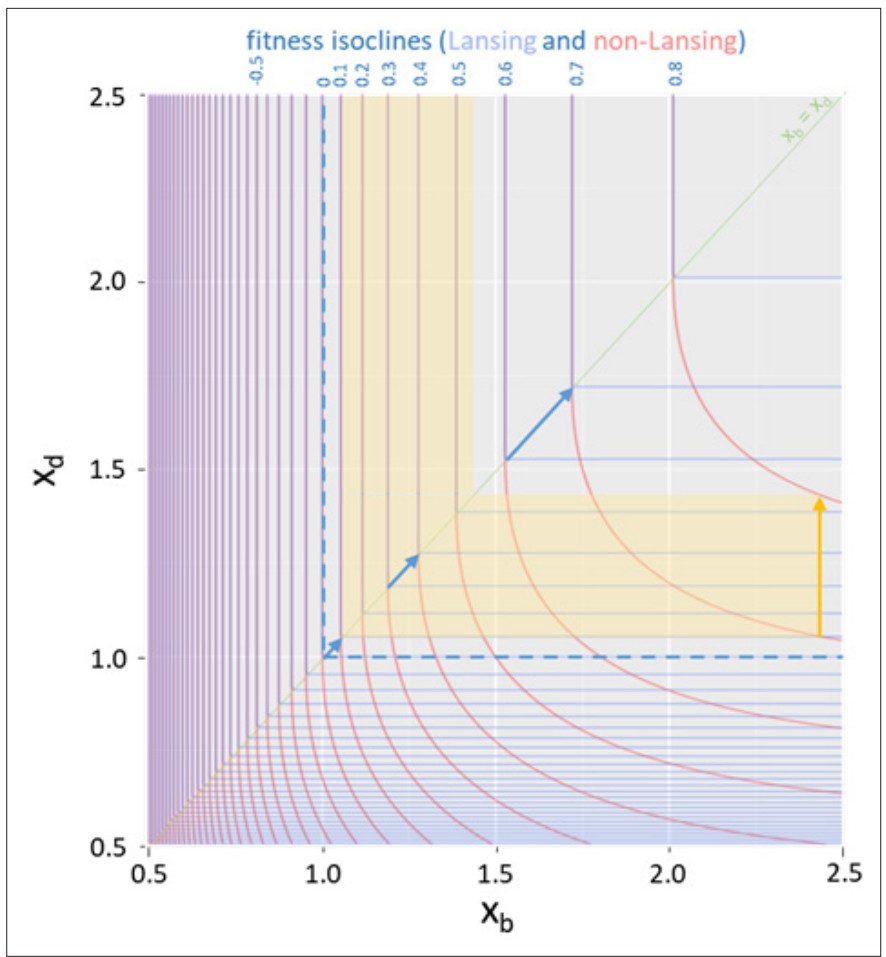

**Figure 5.** The Lansing effect is associated with an increased fitness gradient. We were able to derive Lansing and non-Lansing Malthusian parameters from the model's equations (see Annexe 1–2.3 and 1–5) and plot them as a function of the trait ($x_b$, $x_d$). The diagonal $x_b = x_d$ is drawn in light green. The corresponding isoclines are overlapping above the diagonal but significantly differ below, with non-Lansing fitness (red lines) being higher than that of Lansing's (light blue lines). In addition, the distance between two consecutive isoclines is significantly more important in the lower part of the graph for non-Lansing than Lansing bearing populations. As such, a mutation leading a non-Lansing individual's fitness going from 0.7 to 0.8 (yellow arrow) corresponds to a Lansing individual's fitness going from 0.1 to 0.52. Finally, Hamilton's decreasing force of selection with age can be observed along the diagonal with a growing distance between two consecutive fitness isoclines as $x_b$ and $x_d$ continue increasing.

on the $x_b = x_d$ diagonal. For $x_d > x_b$ (*Figure 5*, upper diagonal), fitness isoclines of the two types of individuals fully overlap, thus showing an equal response of both Lansing and non-Lansing fitness to mutations. In addition, as expected, the fitness of Lansing individuals is equal to that of non-Lansing ones for a given trait. On the lower part of the graph, corresponding to $x_d < x_b$, non-Lansing fitness isoclines separate from that of Lansing individuals, making the fitness of non-Lansing individuals higher to that of Lansing ones for a given trait. Nevertheless, the fitness gradient is significantly stronger for Lansing individuals as represented within *Figure 5* by the yellow arrow and associated yellow area. For an individual of trait ($x_b = 2.45$; $x_d = 1.05$), a mutation making a non-Lansing individual 0.1 in fitness (isocline 0.7 to isocline 0.8) will make a Lansing individual increase its own by 0.42 (isocline 0.1 to above isocline 0.5). With a fourfold difference, the Lansing population produces four times as many individuals as the non-Lansing ones for a given mutation probability. But this reasoning can be extended to any trait ($x_b$, $x_d$) with or without Lansing effect. Organisms ageing rapidly - that is with low $x_b$ and $x_d$ - will see their fitness significantly more affected by a given mutation $h$ than individuals with slower ageing affected by the same mutation. As such, because ageing favors the emergence of genetic variants, ageing populations are therefore more evolvable.

## Discussion

Ageing is, despite its phenotypic diversity (*Jones et al., 2014*), an evolutionarily conserved phenomenon. How ageing evolved, however, is presently debated. Although early theories (*Weismann, 1882*) conceive ageing as adaptive, ageing is now generally viewed as a side-effect, or byproduct, of diminished selective pressure and therefore not adaptive.

The mathematical model we have presented here allows us to propose an alternative theory: ageing necessarily emerges for any system showing the two minimal properties of life (*Trifonov, 2011*), namely (a) reproduction with variation ($x_b$) and (b) organismal maintenance ($x_d$). We formally show that a haploid and asexual organism with these two properties will rapidly evolve, within a few dozen generations, towards a solution such that ($x_b$ - $x_d$) is strictly positive, meaning that the risk of mortality starts to increase before the end of the fertility span. Importantly, the time separating both parameters is independent from their absolute values and only depends on the rate of each, respectively $i_b$ for $x_b$ and $i_d$ for $x_d$. This explains the observed trade-offs (*Kirkwood, 2005*; *Lemaître et al., 2015*; *Rodrigues and Flatt, 2016*) between the fertility of an organism and its lifespan. Thus, our work addresses outstanding questions outlined in the disposable-soma theory (*Kirkwood and Holliday, 1979*) — why and how a highly fertile organism either dies or ages earlier. Indeed, the lower limit condition for the production of descendants by an individual in our model is $x_b * i_b > 1$. As such, an organism with low fertility ($i_b \ll 1$) will obtain a progeny only if fertile longer ($x_b \gg 1$). Conversely, a highly fertile organism will evolve towards its minimum viable condition, requiring only a small $x_d$. The apparent trade-off between fertility and longevity is thus solely a consequence of $x_b * i_b > 1$ and $\lim_{+\infty}(x_b - x_d)_t$. Our model need not implement any constraint on resource allocations or other tradeoffs for this effect to occur.

Because $x_b$ and $x_d$ converge, this favors the onset of a period in which an individual's fertility drops while its risk of dying becomes non-zero; this is the organism entering the 'senescence phase' corresponding to the Smurf phase described in *Rera et al., 2012*. This necessary convergence of fertility's end and senescence's start would thus facilitate the selection of any molecular mechanism that couples the two processes (*Echave, 2021*). Additionally, and in opposition to what is suggested in *Stearns, 1989*, we observe that any two genes that are not functionally linked can be co-selected.

While the Lansing effect somewhat decreases the fitness of individuals within a population, the probability of survival of a population is significantly greater in Lansing populations when in competition with a non-Lansing population of equal Malthusian parameter at $t_0$. We observed, numerically, that this slight increase in survival is mediated by an increase in the genetic variability generated within the population. Thus, we propose that such an active mechanism of ageing can be selected during evolution through its ability to increase an organism's evolvability. As mentioned above, evolvability is understood as the 'the capacity to generate heritable selectable phenotypic variation' (*Kirschner and Gerhart, 1998*). It is an interesting concept as it allows for a trait that has no direct effect on fitness - even a negative one (*Maynard Smith, 1971*) - to be under strong selection, given its ability to generate genetic or phenotypic variation. Furthermore, such a two-phase mechanism would be of great advantage in a constantly varying environment. Indeed, when environmental conditions become less permissive, $x_d$ might be affected and individuals would be pushed to enter the [$x_d$; $x_b$] space earlier, thus increasing the evolvability of the population. This is what we observe in the laboratory where individuals submitted to harsh conditions will enter the Smurf phase earlier than the control conditions (*Rera et al., 2012*). Regarding the nature of the transgenerational effect, our model is agnostic and the mere transmission of any negative effect would be sufficient to exert the function.

Because we, without fail, observe the convergence of the end of fertility and the start of senescence, our generalized model - supported by a formal analysis - predicts a high degree of conservation of ageing, specifically as something that can be selected. This gives rise to organisms that lose homeostatic capacities amidst and during the period of fertility. We have identified a mathematical constraint that explains the biphasic pattern of ageing proposed in *Tricoire and Rera, 2015*, allowing for the positive selection of ageing through evolutionary time. More importantly, the negative impacts of ageing on individuals' fitness seem to be fully compensated at the population level. Our work, at large, thus demonstrates the following: (1) fertility and senescence always converge if an organism is both fertile and homeostatic, (2) ageing populations are more successful through time, and (3) more evolvable. Therefore, we defend that ageing can, in theory, be re-conceptualized as adaptive.

This two phase model is very simple, yet able to describe all types of ageing observed in the wild, including a rapid post-reproductive onset of mortality, a menopause-like mortality plateau, and what we have identified as a two-phase Smurf-like process. The strong mathematical constraint between $x_b$, $x_d$, $i_b$, and $i_d$ limits the possible configurations. Additionally, our mathematical model of ageing, as a two-phase process (*Tricoire and Rera, 2015*), shows that the mortality rate of the second phase of life is considerably constant across *Drosophila* lines of significantly different life expectancies, ranging from 15 to 70 days. In these conditions, if $i_d$ is a constant parameter, can we experimentally affect $x_d$ by acting on $i_b$ and/or $x_d$? Experimental evolution using only *Drosophila* progeny conceived later in the life of the parent has shown that the onset of mortality, within these progeny, occurs rather late, sometimes even after the end of the fertility period (*Burke et al., 2016*; *Rose et al., 2002*). Although the authors report previous studies of their own with divergent results, other independent experiments have led to results suggesting an increase of $x_d$ following an artificial increase of $x_b$ (*Luckinbill and Clare, 1985*; *Sgrò et al., 2000*) as well as the reverse (*Stearns et al., 2000*).

Without the need to implement resource allocation constraints, pleiotropic antagonistic functions nor late-life accumulation of mutations, our model is able to predict the evolution of ageing while encompassing phenomena that previously led to the two above-mentioned theories (mutation accumulation and antagonistic pleiotropy). More importantly, our model suggests a central role of ageing in evolution, as the mathematical constraint we show is likely to apply to any function affecting fertility and homeostasis. Could this broader application of constraints be responsible for the stereotyped gene expression changes - reminiscent of the so-called hallmarks of ageing - we recently described in Smurfs (*Zane et al., 2023*)? Although this model helps us to see the conditions under which ageing is an evolutionarily adaptive force, it is still a toy model. The mortality and fertility functions we used are binary and we are now developing more complex versions of the model, notably to assess the interactions existing between $i_b$, $i_d$, $x_b$, and $x_d$ but more importantly to assess their co-evolution with maturation, sex, ploidy, or varying environmental conditions.

## Materials and methods

See data availability statement for code, packages and the software used.

## Acknowledgements

We thank Dr. Sarah Kaakai for her help in transposing our initial simulation Python codes into the IBMPopSim (*Giorgi et al., 2023*) framework, Dr. Allon Weiner for a few hours of 'naive' discussion that helped explore the interpretations of this model's impact on our perception of ageing, Dr. André Klarsfeld for his numerous useful comments on the manuscript. This work was granted access to the GENCI-sponsored HPC resources of TGCC@CEA under allocation A0090607519. This work has been supported by the Chair 'Modélisation Mathématique et Biodiversité' of Veolia Environnement-Ecole Polytechnique-Museum National d'Histoire Naturelle-Fondation X. SM is funded by the European Union (ERC, SINGER, 101054787). Views and opinions expressed are however those of the author(s) only and do not necessarily reflect those of the European Union or the European Research Council. Neither the European Union nor the granting authority can be held responsible for them. MR is funded by the CNRS. This project was partially funded by the ANR ADAGIO (ANR-20-CE44-0010) and the ATIP/Avenir young group leader program for MR.

## Additional information

### Funding

| Funder | Grant reference number | Author |
| --- | --- | --- |
| Agence Nationale de la Recherche | ANR-20-CE44-0010 | Michael Rera |
| ATIP/Avenir | | Michael Rera |
| TGCC@CEA | A0090607519 | Pierre Jolivet |

| Funder | Grant reference number | Author |
|---|---|---|
| Veolia Environnement - Ecole Polytechnique - Museum d'Histoire Naturelle - Fondation X | Chair "Modélisation Mathématique et Biodiversité" | Sylvie Meleard |
| European Research Council | SINGER 101054787 | Sylvie Meleard |
| Centre National de la Recherche Scientifique | | Michael Rera |

The funders had no role in study design, data collection and interpretation, or the decision to submit the work for publication.

## Author contributions

Tristan Roget, Software, Formal analysis, Investigation; Claire Macmurray, Writing - review and editing; Pierre Jolivet, Software; Sylvie Meleard, Conceptualization, Formal analysis, Supervision, Funding acquisition, Investigation, Methodology, Writing - original draft; Michael Rera, Conceptualization, Resources, Data curation, Software, Supervision, Funding acquisition, Investigation, Methodology, Writing - original draft, Project administration, Writing - review and editing

## Author ORCIDs

Michael Rera ⓘ https://orcid.org/0000-0002-6574-6511

Joint public review: https://doi.org/10.7554/eLife.92914.3.sa1
Author response https://doi.org/10.7554/eLife.92914.3.sa2

# Additional files

## Supplementary files

- MDAR checklist

## Data availability

All data are available for download at https://github.com/MichaelRera/EvoAgeing/tree/main/article_sims. Package IBMPopSim (R package IBMPopSim v0.3.1): https://cran.r-project.org/web/packages/IBMPopSim/index.html. Github repository for simulation results and code: https://github.com/MichaelRera/EvoAgeing/tree/main/article_sims (copy archived at *Rera, 2024*). Environment for simulations using IBMPopSim: https://mybinder.org/v2/gh/MichaelRera/EvoAgeing/HEAD. Exploring parameters for Lansing populations: Modele_Lansing_evo.ipynb. Exploring parameters for non-Lansing populations Modele_nonLansing_evo.ipynb. Lansing / non-Lansing competition for equal Malthusian parameters L_nL_compet_eqMalth.ipynb. Lansing / non-Lansing competition (xb-xd) € [-10; 10] L_nL_compet_heteroPop.ipynb.

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

## Appendix 1

### 1 The mathematical individual-based bd model

We model an haploid and asexual population of individuals with evolving life-histories by a stochastic individual-based model, similar to the one introduced in *Méléard et al., 2019* and a particular case of *Ferrière and Tran, 2009*. Each individual is characterized by its age and by a life-history trait $x = (x_b, x_d) \in \mathbb{R}_+^2$ that describes for each individual the age $x_b$ at the end of reproduction and the age $x_d$ when mortality becomes positive. The trait can change through time, by mutations occuring continuously in time.

More precisely, the Markovian dynamics of the population process is defined as follows. The individuals reproduce and die independently. An individual with trait $(x_b, x_d)$ reproduces at rate $i_b$ as long as it is younger than $x_b$. Further, he cannot die as long as it is younger than $x_d$ and has a natural death rate $i_d$ after age $x_d$.

The life-history of an individual with trait $x = (x_b, x_d)$ is described by the couple of birth and death functions $(B_x, D_x^c)$ defined on $\mathbb{R}_+$ by

$$\forall a \in \mathbb{R}_+, \qquad B_x(a) = i_b \mathbb{1}_{a \leq x_b}, \quad D_x^c(a) = i_d \mathbb{1}_{a > x_d} + cN. \tag{1}$$

Here, the individual age $a$ is the physical age, $N$ the (varying) population size and $c > 0$ the competition pressure exerted by an individual on another one. The death rate will be extended to

$$D_x^c(0) = +\infty, \text{ for } x_b < 0 \text{ or } x_d > 0,$$

meaning that an individual appearing by mutation will be able to survive only if the two components if its trait are non negative.

Note that the date of birth and lifespan of an individual are stochastic and the law of the lifespan on an individual with trait $x$ born at time $\tau$ is given by $f_x(s) = D_x^c(s) \exp(- \int_\tau^{\tau+s} D_x^c(a)da)$.

We also take into account genetic mutations which create phenotypic variation, and which added to competition between individuals, will lead to natural selection.

At each reproduction event, a mutation appears instantaneously on each trait $x_b$ and $x_d$ independently with probability $p \in ]0, 1[$. If the trait $x_b$ mutates (resp. if $x_d$ mutates), the trait of the newborn is $x_b + h_b$ (resp. $x_d + h_d$). The mutation effect $h_b$ (resp. $h_d$) is distributed following a centered Gaussian law with variance $\sigma^2$. This Gaussian law is denoted by $k(h)dh$.

Note that a similar model has been defined in *Méléard et al., 2019*, including a Lansing effect on the reproductive lineage of "old" individuals.

### 2 The Malthusian parameter

#### 2.1 The demographic parameters

We now introduce the classical demographic parameters for age-structured (without competition) population, where all individuals have the same trait $x \in \mathbb{R}_+^2$ (*Charlesworth, 1994*). We are looking for a triplet $(\lambda(x), N_x, \phi_x)$ where $\lambda(x) \in \mathbb{R}$ is the Malthusian parameter, $N_x(a), a \in \mathbb{R}_+$ the stable age distribution and $\phi_x(a), a \in \mathbb{R}_+$ the reproductive value. They describe the asymptotic growth of the population dynamics and measure the fitness of life-histories: $\lambda(x)$ is the growth rate of the population at its demographic equilibrium, $N_x$ the age distribution of the population and $\phi_x(a)$ is the probability that an individual with trait $x$ has a newborn after age $a$. It is known (*Charlesworth, 1994*), that $(\lambda(x), N_x, \phi_x)$ is solution of the direct and dual eigenvalue problems:

$$\begin{cases} -\partial_a N_x(a) - D_x(a)N_x(a) = \lambda(x)N_x(a) \\ N_x(0) = \int_0^{+\infty} B_x(\alpha)N_x(\alpha)d\alpha, \quad N_x(0) = 1, \end{cases} \tag{2}$$

$$\begin{cases} \partial_a \phi_x(a) - D_x(a)\phi_x(a) + B_x(a)\phi_x(0) = \lambda(x)\phi_x(a) \\ \phi_x(0) = 1 \end{cases} \tag{3}$$

where $B_x(a) = i_b \mathbb{1}_{\{a > x_b\}}$ and $D_x(a) = i_d \mathbb{1}_{\{a > x_d\}}$.

### Proposition 2.1

For all $x \in \mathbb{R}_+^2$, there exists a unique solution $(\lambda(x), N_x, \phi_x) \in \mathbb{R} \times L^1(\mathbb{R}_+) \times L^\infty(\mathbb{R}_+)$ of (2) and (3). The Malthusian parameter $\lambda(x)$ is the unique solution of the equation:

$$i_b \int_0^{x_b} e^{-i_d(a-x_d)_+ - \lambda(x)a} da = 1. \tag{4}$$

The stable age distribution $N_x$ and the reproductive value $\phi_x$ verify

$$N_x(a) = e^{-i_d(a-x_d)_+ - \lambda(x)a}, \quad \phi_x(a) = \frac{i_b \mathbb{1}_{a \leq x_b}}{N_x(a)} \int_a^{x_b} N_x(\alpha) d\alpha. \tag{5}$$

### Proof

The proof is straightforward by solving the first equations in (2) and (3), and then by using the equations satisfied by the boundary conditions. □

### Remark 2.2

The quantities $\lambda(x), N_x, \phi_x$ are the eigenelements (Proposition 2.1) associated with the linear operator that generates the dynamics $v_x(t, a)$ of a non density dependent population with age structure and birth-death rates given by $(B_x, D_x)$. More precisely, $v_x(t, a)$ satisfies the McKendrick Von-Foerster Equation

$$\begin{cases} \partial_t v_x(t, a) + \partial_a v_x(t, a) = -D_x(a)v_x(t, a), & t \geq 0, a \geq 0 \\ v_x(t, 0) = \int_{\mathbb{R}_+} B_x(\alpha)v_x(t, \alpha). \end{cases}$$

The use of these quantities as an indicator of fitness is justified by the convergence of $e^{-\lambda(x)t} v_x(t, a)$ to $(\int_{\mathbb{R}_+} v_x(0, \alpha)\phi_x(\alpha)d\alpha)N_x(a)$ as $t$ tends to infinity (**Perthame, 2007** for example).

## 2.2 Computation and regularity of the Malthusian parameter

The Malthusian parameter $\lambda(x)$ is defined as the unique real number such that

$$i_b \int_0^{x_b} e^{-i_d(a-x_d)_+ - \lambda(x)a} da = 1.$$

Let us introduce

$$U_1 = \{x \in \mathbb{R}_+^2 : x_b < x_d\}, \ U_2 = \{x \in \mathbb{R}_+^2 : x_d < x_b\}, \ \mathcal{H} = \{x \in \mathbb{R}_+^2 : x_b = x_d\}. \tag{6}$$

For all $x \in U_1 \cup \mathcal{H}$, the Malthusian parameter $\lambda(x)$ satisfies:

$$i_b \int_0^{x_b} e^{-\lambda(x)a} da = 1 = \frac{i_b}{\lambda(x)}(1 - e^{-x_b \lambda(x)}).$$

Then $\lambda(x)$ can be numerically computed by Newton's method applied to the function $K_{x_b}(\lambda) = \frac{1}{\lambda}(1 - e^{-x_b \lambda}) - \frac{1}{i_b}$, since $\lambda(x)$ is solution of $K_{x_b}(\lambda) = 0$, .

In the case where $x \in U_2$, we have

$$i_b \int_0^{x_b} e^{-i_d(a-x_d)_+ - \lambda(x)a} da = i_b \int_0^{x_d} e^{-\lambda(x)a} da + i_b \int_{x_d}^{x_b} e^{-i_d(a-x_d) - \lambda(x)a} da$$

$$= i_b \left\{ \frac{1}{\lambda(x)}(1 - e^{-\lambda(x)x_d}) + \frac{e^{i_d x_d}}{\lambda(x) + i_d} \left( e^{-(\lambda(x)+i_d)x_d} - e^{-(\lambda(x)+i_d)x_b} \right) \right\},$$

which has to be equal to 1. That involves a function

$$H_{(x_b, x_d)}(\lambda) = \frac{1}{\lambda}(1 - e^{-\lambda_d}) + \frac{e^{i_d x_d}}{\lambda + i_d} \left( e^{-(\lambda+i_d)x_d} - e^{-(\lambda+i_d)x_b} \right) - \frac{1}{i_b}.$$

Newton's method still allows to resolve numerically the equation and find $\lambda(x)$.

Let us now prove some regularity properties of the Malthusian parameter. We show that its gradient is a simple function of the stable age distribution, the reproductive value and the mean generation time $G$ defined for all $x$ by

$$G(x) = i_b \int_0^{x_b} a N_x(a) da.$$

## Proposition 2.3

The function $x \in (\mathbb{R}_+^*)^2 \mapsto \lambda(x)$ is of class $\mathcal{C}^1$ and we have:

$$\forall x \in (\mathbb{R}_+^*)^2, \qquad \nabla \lambda(x) = \frac{1}{G(x)} \left( i_b N_x(x_b), i_d N_x(x_d) \phi_x(x_d) \right).$$

Note that the derivatives are positive, meaning that $x_b \to \lambda(x)$ and $x_d \to \lambda(x)$ are non decreasing.

## Proof

Coming back to the definition of $\lambda$ and using the the implicit function theorem, we obtain that $\lambda$ is differentiable and

$$
\begin{aligned}
x \in U_1, \quad \frac{\partial \lambda(x)}{\partial x_b} &= \frac{i_b e^{-\lambda(x) x_b}}{G(x)} = \frac{i_b N_x(x_b)}{G(x)} \quad ; \quad \frac{\partial \lambda(x)}{\partial x_d} = 0 = \frac{N_x(x_d) \phi_x(x_d)}{G(x)}; \\
\forall x \in U_2, \quad \frac{\partial \lambda(x)}{\partial x_b} &= \frac{i_b e^{-i_d(x_b - x_d)} e^{-\lambda(x) x_b}}{G(x)} = \frac{i_b N_x(x_b)}{G(x)} \\
\frac{\partial \lambda(x)}{\partial x_d} &= \frac{i_b i_d e^{i_d x_d} \int_{x_d}^{x_b} e^{-(i_d + \lambda(x)) a} da}{G(x)} = \frac{i_b i_d N_x(x_d) \phi_x(x_d)}{G(x)}.
\end{aligned}
\tag{7}
$$

We deduce that $\lambda$ has continuous partial derivatives, which concludes the proof. $\square$

## 2.3 Viability set

The viability set is the set $\mathcal{V} \subset \mathbb{R}_+^2$ of traits $x = (x_b, x_d)$ such that $\lambda(x) > 0$. From **Equation 4**, $\lambda(x) > 0$ if and only if the mean number $R(x_b, x_d)$ of descendants per individual is larger than one, i.e if and only if we have:

$$R(x_b, x_d) := i_b \int_0^{x_b} e^{-i_d(a - x_d)_+} da > 1. \tag{8}$$

A precise characterization of the set $\mathcal{V}$ is given in Lemma 2.4. In **Figure 1**, we represent the set $\mathcal{V}$ for $i_b = 1.5$ and $i_d = 2$.

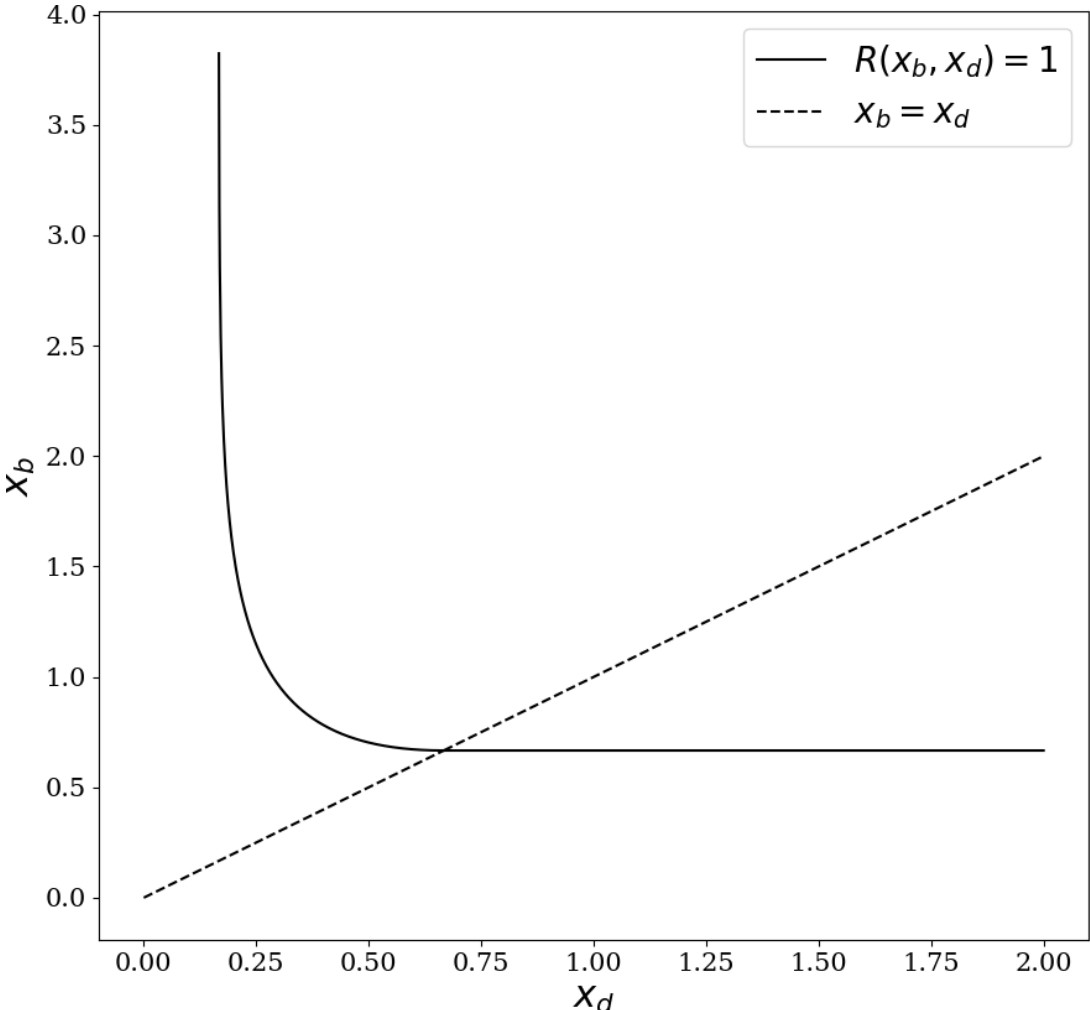

**Appendix 1—figure 1.** The set $\mathcal{V} = \{(x_b, x_d) \in \mathbb{R}_+^2 ; R(x_b, x_d) > 1\}$ is the convex set delimited by the black curve with equation $R(x_b, x_d) = 1$.

## Lemma 2.4
We have:

$$\mathcal{V} = \{x \in \mathbb{R}_+^2 ; x_b > x_d - \log(i_d x_d + 1 - (i_b/i_d)) \text{ if } x_b > x_d ; i_b x_b > 1 \text{ if } x_b \leq x_d\},$$

and for all $x \in \mathcal{V}$, $\lambda(x) \leq i_b$. Moreover, the map $x \in \mathcal{V} \mapsto \nabla\lambda(x)$ is Lipschitz continuous.

## Proof
We are looking for which $x = (x_b, x_d) \in \mathbb{R}_+^2$, the mean number of descendants $R(x_b, x_d)$ is greater than 1. Recall that $R(x) = i_b \int_0^{x_b} \exp\left(-i_d \int_0^a \mathbb{1}_{\alpha > x_d} d\alpha\right) da$. For $x \in U_1$ (defined in **Equation 6**), we have $R(x) = i_b x_b$ and $R(x) > 1$ if and only if $i_b x_b > 1$. For $x \in U_2$, we have $R(x) = i_b x_d + (i_b/i_d) - (i_b/i_d)e^{-i_d(x_b - x_d)}$ and $R(x) > 1$ if and only if $x_b > x_d - \log(i_d x_d + 1 - (i_b/i_d))$. We conclude for the first assertion arguing that the map $\lambda \mapsto i_b \int_0^{x_b} \exp\left(-i_d \int_0^a \mathbb{1}_{\alpha > x_d} i_d \alpha - \lambda a\right) da$ is decreasing. Let us now show that $\lambda(x)$ is upper-bounded by $i_b$. Assume that there exists $x \in \mathcal{V}$ such that $\lambda(x) > i_b$. Then

$$1 = i_b \int_0^{x_b} e^{-i_d(a - x_d)_+ - \lambda(x)a} da < i_b \int_0^{x_b} e^{-i_b a} da = 1 - e^{-x_b},$$

which is absurd and allows us to conclude. The next claim is shown arguing that the map $x \in \mathcal{V} \mapsto \nabla\lambda(x)$ is differentiable on $U_1 \cup U_2$ and admits bounded partial derivatives. □

Let us develop different examples:

In the case where $i_b = 0.01$ and $i_d = 1$, we obtain

$$R(x_b, x_d) > 1 \iff x_d + \left(1 - \left(2.01\right)^{-\frac{1}{1.01}}\right) > 100,$$

which gives essentially that $x_d$ has to be greater than 100.

In the case where $i_b = i_d$, the formula is simpler. We obtain

$$R(x_b, x_d) > 1 \iff x_d + \frac{1}{i_b}\left(1 - 3^{-\frac{1}{2}}\right) > \frac{1}{i_b}.$$

We deduce

$$R(x_b, x_d) > 1 \iff x_d > \frac{1}{i_b\sqrt{3}}.$$

If we assume that $i_b = i_d = 1$ then we obtain that

$$R(x_b, x_d) > 1 \iff x_d > \frac{1}{\sqrt{3}} = 0.577.$$

Let us finally note that if we assume to be in the limit of the canonical equation and then to be in the case when $x_b - x_d = \frac{\log 3}{2 i_b}$, we also obtain a characterization of the viability set using $x_b$:

$$R(x_b, x_d) > 1 \iff x_b > \frac{1}{i_b}\left(\frac{1}{\sqrt{3}} + \frac{\log 3}{2}\right).$$

For $i_b = 1$, that gives $x_b > 1.126$

## 3 Monomorphic equilibrium

Let us come back to the general case with competition, but for a monomorphic population with trait $x$ (and then without mutation). It can be proved (**Méléard and Tran, 2009** Proposition 2.4) that for a large population, the stochastic process converges in probability to the solution of the following Gurtin-MacCamy partial differential equation (see **Gurtin and Maccamy, 1974**).

$$\begin{cases} \partial_t n_x(t, a) + \partial_a n_x(t, a) = -\left(D_x(a) + c\int_{\mathbb{R}_+} n_x(t, \alpha)d\alpha\right)n_x(t, a) \\ n_x(t, 0) = \int_0^{+\infty} B_x(\alpha)n_x(t, \alpha)d\alpha, \quad (t, a) \in \mathbb{R}_+^2. \end{cases} \tag{9}$$

This equation describes the density-dependent dynamics of a large population with trait $x$ (without mutation). The trait $x \in \mathbb{R}_+^2$ being given, let us study the positive equilibria of the equation

For $x \in \mathcal{V}$, **Equation 9** admits a unique non-trivial solution:

### Proposition 3.1

For all $x \in \mathcal{V}$, there exists a unique globally stable equilibrium $\bar{n}_x$ to **Equation 9**, that is a solution of

$$\begin{cases} -\partial_a \bar{n}_x(a) - \left(D_x(a) + c\int_{\mathbb{R}_+} \bar{n}_x(\alpha)d\alpha\right)\bar{n}_x(a) = 0 \\ \bar{n}_x(0) = \int_0^{+\infty} B_x(\alpha)\bar{n}_x(\alpha)d\alpha, \end{cases} \tag{10}$$

which satisfies $\lambda(x) = c\int_{\mathbb{R}_+} \bar{n}_x(\alpha)d\alpha$.

Note that

$$\bar{n}_x(0) = \frac{\lambda(x)}{c\int_0^{+\infty} N_x(\alpha)d\alpha}.$$

## Proof

The existence part of the proof is trivial from *Cansell et al., 2023* and Proposition 2.1 using that $\mathcal{V} = \{x \in \mathbb{R}_+^2 : \lambda(x) > 0\}$. The long-time behavior of the solutions of (9) is studied in [*Webb, 1985*, Section 5.4]. □

## 4 Canonical equation

### 4.1 Invasion fitness

We now compute the invasion fitness function associated with the individual-based model. We use the definition of invasion fitness given in *Méléard and Tran, 2009*. The invasion fitness $1 - z(y, x)$ of a mutant with trait $y$ in a resident population with trait $x$ is defined as the survival probability of an age-structured branching process with birth rates $B_x(a)$ and death rates $D_x(a) + c \int_{\mathbb{R}_+} \bar{n}_x(a) da$.

### Proposition 4.1

Let $y \in (\mathbb{R}_+^*)^2$ and $x \in \mathcal{V}$, we have

$$1 - z(y, x) = \left[ \frac{\lambda(y) - \lambda(x)}{i_b} \right]_+.$$

### Proof

The proof is a direct application of Equation (3.6) in *Méléard and Tran, 2009*. □

### 4.2 Trait Substitution sequence and Canonical equation

For this part, we refer principally to *Méléard and Tran, 2009* where the Theory of Adaptive Dynamics is rigorously developed for general age-structured populations.

We introduce the canonical equation describing the evolution of the trait $x = (x_b, x_d)$ at a mutation time-scale, under the assumptions of adaptive dynamics (large population, rare and small mutation, invasion and fixation principle, as well known since Metz et al., Dieckman-Law). In *Méléard and Tran, 2009*, it is shown that this equation can be obtained as a two-step limit from the individual based model. The first step consists in defining the Trait Substitution Process describing the successive advantageous mutant invasions in monomorphic populations at equilibrium. It is obtained as support dynamics of the measure-valued limit of the rescaled population process (at the mutation time-scale), when mutations are rare (but not small). The measure-valued limiting process is rigorously derived from the individual-based model in *Dambroise et al., 2016* Section 3. It jumps from a state $\delta_x(dz) n_x(a) da$ to a state $\delta_y(dz) - n_y(a) da$. The trait support process takes values in $\mathcal{V}$ and its dynamics is described as follows.

### Definition 4.2

The Trait Substitution Sequence is the càdlàg process $(X_t, t \geq 0)$ with values in $\mathcal{V}$ whose law is characterized by the infinitesimal generator $L$ defined for all bounded and measurable function $\varphi : \mathcal{V} \to \mathbb{R}$ by:

$$L\varphi(x) = \int_{\mathbb{R}^2} (\varphi(x + (h_1, h_2)) - \varphi(x)) \left[ \frac{\lambda(x + (h_1, h_2)) - \lambda(x)}{i_b} \right]_+ \frac{\lambda(x)}{c \int_0^{+\infty} N_x(a) da} \mu(dh_1, dh_2),$$

where $\mu(dh) = \frac{\delta_0(dh_2) k(h_1) dh_1 + \delta_0(dh_1) k(h_2) dh_2}{2}$ and the distribution $k$ has been defined in Section 1.

Note that since by Proposition 2.3, the partial derivatives of $\lambda$ are positive, then the increment $\lambda(x + (h_1, h_2)) - \lambda(x)$ is non negative if and only if $h_1$ and $h_2$ are non negative.

The second step consists in assuming that mutation amplitudes are small and of order $\epsilon$, for $\epsilon > 0$. We then define the rescaled process $X^\epsilon$ by $X^\epsilon(t) = \epsilon X(\frac{t}{\epsilon^2})$. introduce the Canonical Equation that describes the limit behaviour of the Trait Substitution Sequence when mutations are small.

### Proposition 4.3

Let $T > 0$. Assume that $X^\epsilon(0)$ converges to $x^0 \in \mathcal{V}$ in probability. Then the sequence of processes $(X^\epsilon)_\epsilon$ converges in law in the Skorohod space $\mathbb{D}([0, T], \mathcal{V})$ to the solution $(x(t), t \geq 0)$ of the ordinary differential equation:

$$\frac{dx}{dt} = \frac{\lambda(x)}{4c \int_0^{+\infty} N_x(\alpha)d\alpha} \frac{\nabla\lambda(x)\sigma^2}{i_b}, \quad x \in \mathcal{V} \subset \mathbb{R}_+^2 \tag{11}$$

Recall that the Malthusian parameter $\lambda(x)$ is defined in *Fabian, 2011*, the stable age distribution $N_x$ is defined in *Frenk and Houseley, 2018* and $\sigma^2(x)$ denotes the variance of the mutation kernel. Recall that (see Proposition 2.3)

$$\nabla\lambda(x) = \frac{1}{G(x)} \left( i_b N_x(x_b), i_d N_x(x_d)\phi_x(x_d) \right). \tag{12}$$

It describes the strength of selection at ages $x_b$ and $x_d$. Hence, this canonical equation allows to interpret the age specific strength of selection at ages $x_b$ and $x_d$ as the evolution speed of the traits $x_b$ and $x_d$ respectively, under the assumptions of adaptive dynamics.

### Proof
The proof is classical and can be easily adapted from that of [6, Theorem 4.1]. The canonical equation only charges the set $\mathcal{V} \subset \mathbb{R}_+^2$ (defined in Section 2.1) and writes as follows:

$$\frac{dx}{dt} = -\nabla_y z(x, x) \frac{\bar{n}_x(0)}{2} \int_{\mathbb{R}_+} h^2 k(h)dh, \quad x \in \mathcal{V}. \tag{13}$$

The set $\mathcal{V}$ is the set of traits that admit a positive stable monomorphic equilibrium $\bar{n}_x$ in a such way that $\bar{n}_x(0)$ equals the birth rate of a mutant (see Proposition 3.1); $\sigma^2$ is the variance of the mutations and $1 - z(y, x)$ is the invasion fitness. Computing these parameters gives (*Equation 11*). □

In *Figure 2*, we present a simulation of a solution of *Equation 11*. We observe that the traits $x_b$ and $x_d$ increase with time (*Figure 2a, b*), with decreasing speed tending to zero. The trait $x_b(t) - x_d(t)$ converges to some positive number (*Figure 2c*) that we can rigorously compute. That is the aim of the next section.

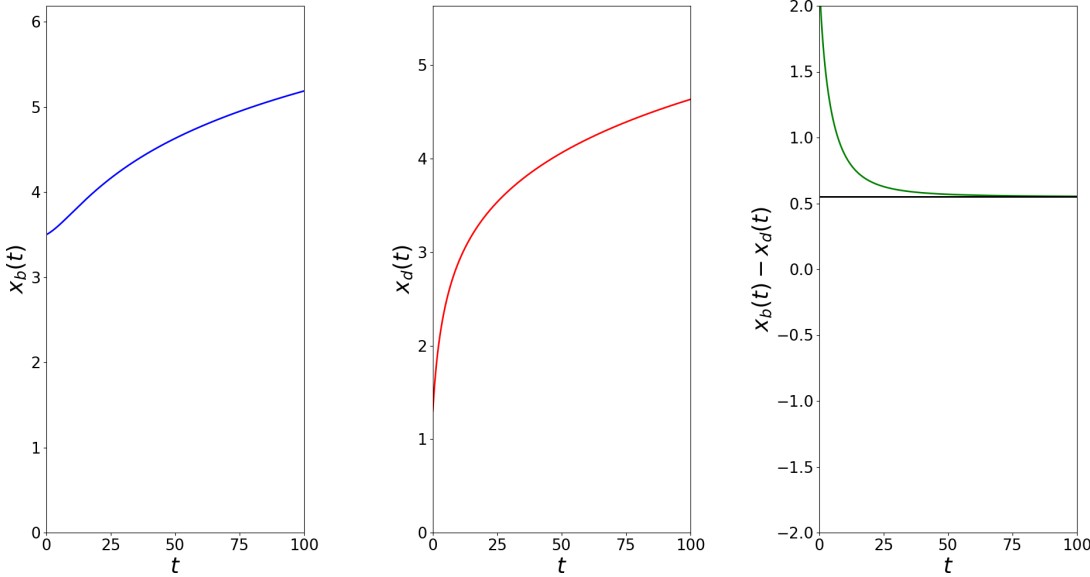

**Appendix 1—figure 2.** Simulation of the canonical equation with $x^0 = (3.5, 1.3)$ and $i_b = i_d = 1$. (**a**) Dynamics of $x_b$. (**b**) Dynamics of $x_d$. (**c**) Dynamics of, $x_b - x_d$ the black curve has equation. $y = \log(3)/2$.

## 4.3 Long-time behaviour of the canonical equation
In this section we study the long-time behaviour of the solutions of the Canonical *Equation 11*. We prove the following theorem.

### Theorem 4.4
Let $x^0 \in \mathcal{V}$ and let $(x(t), t \geq 0)$ be the solution of (11) started at $x^0 \in \mathcal{V}$. Then we have:

$$x_b(t) - x_d(t) \xrightarrow[t \to +\infty]{} \frac{\log(1 + \frac{i_b + i_d}{i_d})}{i_b + i_d}.$$

We first prove the following lemma. We always denote $U_1 = \{x \in \mathcal{V} : x_b < x_d\}$, $U_2 = \{x \in \mathcal{V} : x_d < x_b\}$ and $\mathcal{H} = \{x \in \mathcal{V} : x_d = x_b\}$.

## Lemma 4.5

Let $x^0 \in \mathcal{V}$ and let $(x(t), t \geq 0)$ be the solution of **Equation 11** started at $x^0 \in \mathcal{V}$. Then we have:

- (i) *There exists $T > 0$, such that for all $t \geq T$, $x(t) \in U_2 \cup \mathcal{H}$.*
- (ii) *There exists $C > 0$ such that for all $t \geq 0$, $|x_b(t) - x_d(t)| < C$,*
- (iii) *We have $x_b(t)$ increases to $+\infty$, $x_d(t)$ increases to $+\infty$ and $\lambda(x(t)) \to i_b$ as $t \to +\infty$.*

## Proof

For all $x \in \mathcal{V}$, let us define:

$$v(x) = \frac{\lambda(x) \int_0^{+\infty} h^2 k(h) dh}{2 i_b c G(x) \int_0^{+\infty} N_x(\alpha) d\alpha}.$$

and we remark that there exist $\underline{v}(x^0), \overline{v}(x^0) > 0$ such that $\underline{v}(x^0) \leq v(x) \leq \overline{v}(x^0)$. (i): Let $T := \inf\{t \geq 0 : x(t) \in U_2 \cup \mathcal{H}\} \in [0, +\infty]$. We first show that $T < +\infty$. If $x^0 \in U_2 \cup \mathcal{H}$, it is obvious. If $x^0 \in U_1$, assume that $T = +\infty$. Then for all $t \geq 0$, $x_d(t) = x_d^0$. Indeed, as soon as $x_b < x_d$, $\phi(x_d) = 0$ and the trait $x_d$ does not move (see **Equation 7**). We obtain that

$$\forall t \geq 0, \quad \frac{d(x_b(t) - x_d(t))}{dt} = \frac{dx_b(t)}{dt} \geq v_1(x) i_b e^{-\lambda(x) x_b} \geq \underline{v}(x^0) i_b e^{-\lambda(x) x_b} > 0,$$

that allows us to obtain the contradiction. So we have $T < +\infty$. We conclude the proof arguing that for all $t \geq 0$ such that $x(t) \in \mathcal{H}$, $dx_d(t)/dt = 0$ and $dx_b(t)/dt > 0$. (ii): By (i), we assume without loss of generality that $(x(t), t \geq 0) \subset U_2 \cup \mathcal{H}$. By **Equation 11**, we obtain that:

$$\begin{aligned}
\frac{d(x_b(t) - x_d(t))}{dt} &= i_b v(x(t)) \left( e^{-i_d(x_b(t) - x_d(t)) - \lambda(x)(t) x_b(t)} - i_d \int_{x_d(t)}^{x_b(t)} e^{-i_d(a - x_d(t)) - \lambda(x)(t) a} da \right) \\
&= i_b v(x(t)) \left( e^{-i_d(x_b(t) - x_d(t)) - \lambda(x)(t) x_b(t)} \right. \\
&\qquad \left. - \frac{i_d e^{i_d x_d(t)}}{i_d + \lambda(x(t))} (e^{-(i_d + \lambda(x(t))) x_d(t)} - e^{-(i_d + \lambda(x(t))) x_b(t)}) \right) \\
&= \frac{i_b v(x(t))}{i_d + \lambda(x(t))} \left( (2 i_d + \lambda(x(t))) e^{-i_d(x_b(t) - x_d(t)) - \lambda(x)(t) x_b(t)} - i_d e^{-\lambda(x)(t) x_d(t)} \right) \\
&= \frac{i_b i_d v(x(t)) e^{-\lambda(x(t)) x_d(t)}}{i_d + \lambda(x(t))} \left( \frac{2 i_d + \lambda(x(t))}{i_d} e^{-(i_d + \lambda(x(t)))(x_b(t) - x_d(t))} - 1 \right).
\end{aligned} \qquad (14)$$

By **Equation 14** and using the fact that for $x \in \mathcal{V}$, $0 < \lambda(x) \leq i_b$ (Lemma 2.4), we obtain that

$$\frac{d(x_b(t) - x_d(t))}{dt} \leq \frac{i_b i_d v(x(t)) e^{-\lambda(x(t)) x_d(t)}}{i_d + \lambda(x(t))} \left( \frac{2 i_d + i_b}{i_d} e^{-i_d(x_b(t) - x_d(t))} - 1 \right).$$

From the previous inequality, we deduce that on the set

$$\left\{ t \geq 0 : x_b(t) - x_d(t) > \frac{\log(\frac{2 i_d + i_b}{i_d})}{i_d} \right\},$$

the quantity $x_b(t) - x_d(t)$ is decreasing, which allows us to conclude.

(iii): As before and by (i), we assume without loss of generality that $(x(t), t \geq 0) \subset U_2 \cup \mathcal{H}$. Using (ii) and since $\lambda(x) \leq i_b$ (Lemma 2.4), we obtain that

$$\frac{\mathrm{d}x_b(t)}{\mathrm{d}t} = i_b v(x(t))e^{-i_d(x_b(t)-x_d(t))}e^{-\lambda(x(t))x_b(t)}$$

$$\geq i_b \underline{v}(x^0)e^{-C}e^{-i_b x_b(t)},$$

that allows to conclude that $x_b(t)$ increases to $+\infty$ and by (ii) we also have a similar behavior for $x_d(t)$. We now prove that $\lambda(x(t)) \to i_b$ as $t \to +\infty$. Let us recall that for all $t \geq 0$, $\lambda(x(t))$ is the unique solution of

$$i_b \int_0^{x_b(t)} e^{-i_d(a-x_d(t))_+ - \lambda(x(t))a} \, da = 1,$$

that we rewrite

$$i_b \int_0^{x_d(t)} e^{-\lambda(x(t))a} \, da + i_b \frac{(e^{-\lambda(x(t))x_d(t)} - e^{-i_d(x_b(t)-x_d(t)) - \lambda(x(t))x_b(t)})}{i_d + \lambda(x(t))} = 1. \tag{15}$$

The map $t \mapsto \lambda(x(t))$ is clearly increasing (using *Equation 7* and the positivity of $x_b'(t)$ and $x_d'(t)$) and bounded by $i_b$. So there exists $\lambda^* > 0$ such that $\lambda(x(t)) \to \lambda^*$. By taking the limit $t \to +\infty$ in *Equation 15* and using the previous part of the proof, we deduce that

$$i_b \int_0^{+\infty} e^{-\lambda^* a} \, da = 1,$$

and $\lambda^* = i_b$ that concludes the proof. $\square$

We now prove Theorem 4.4.

## Proof of Theorem 4.4

By Lemma 4.5 (i), we assume without loss of generality that $(x(t), t \geq 0) \subset U_2 \cup \mathcal{H}$, i.e that for all $t \geq 0$, $x_b(t) - x_d(t) \geq 0$. We recall that Equality *Equation 14* gives:

$$\frac{\mathrm{d}x_b(t) - x_d(t)}{\mathrm{d}t} = \frac{i_b i_d v(x(t))e^{-\lambda(x(t))x_d(t)}}{i_d + \lambda(x(t))} \left( \frac{2i_d + \lambda(x(t))}{i_d} e^{-(i_d + \lambda(x(t))(x_b(t)-x_d(t)))} - 1 \right). \tag{16}$$

We define $f, h : \mathbb{R}_+ \to \mathbb{R}$ by:

$$f(t) = \frac{i_b i_d v(x(t))e^{-\lambda(x(t))x_d(t)}}{i_d + \lambda(x(t))}$$

and

$$h(t) = \frac{2i_d + \lambda(x(t))}{i_d} e^{-(i_d + \lambda(x(t)))(x_b(t)-x_d(t))} - \frac{2i_d + i_b}{i_d} e^{-(i_d + i_b)(x_b(t)-x_d(t))}.$$

Note that $h(t) \to 0$ as $t \to +\infty$ using Lemma 4.5 (ii). Let us also define $u(t) = x_b(t) - x_d(t)$. So *Equation 16* rewrites

$$\frac{\mathrm{d}u(t)}{\mathrm{d}t} = f(t) \left( \frac{2i_d + i_b}{i_d} e^{-(i_d + i_b)u(t)} - 1 + h(t) \right).$$

We deduce that for all $\epsilon > 0$, there exists $t_0 > 0$ such that for all $t \geq t_0$,

$$f(t) \left( \frac{2i_d + i_b}{i_d} e^{-2(i_b + i_d)u(t)} - 1 - \epsilon \right) \leq \frac{\mathrm{d}u(t)}{\mathrm{d}t} \leq f(t) \left( \frac{2i_d + i_b}{i_d} e^{-(i_d + i_b)u(t)} - 1 + \epsilon \right). \tag{17}$$

Let us consider the differential equation

$$\frac{\mathrm{d}w(t)}{\mathrm{d}t} = f(t) \left( \frac{2i_d + i_b}{i_d} e^{-(i_b + i_d)w(t)} - 1 + \epsilon \right).$$

By using the change of variables $s = e^{(i_b + i_d)w}$, we solve the previous equation and we find that there exists a constant $C(x^0)$ such that

$$w(t) = \frac{1}{i_b + i_d} \log\left(\frac{2i_d + i_b}{i_d(1-\epsilon)} - \frac{C(x^0)}{1-\epsilon} \exp\left(-(i_b + i_d)(1-\epsilon)\int_0^t f(s)ds\right)\right).$$

We conclude by proving that the integral above tends to infinity as $t$ tends to infinity. First, the inequality $x_b(t) \geq x_d(t)$ implies that

$$f(t) \geq \frac{i_b i_d v(x(t))}{i_d + \lambda(x(t))} e^{-\lambda(x(t))x_b(t)}.$$

Moreover, *Equation 11* gives that

$$v(x(t))e^{-\lambda(x(t))x_b(t)} = e^{i_d(x_b(t) - x_d(t))} x_b'(t).$$

Since $\lambda(x(t)) \leq i_b$, we obtain that

$$f(t) \geq \frac{i_b i_d}{i_b + i_d} x_b'(t)$$

and that

$$\int_0^t f(s)ds \geq \frac{i_b i_d}{i_b + i_d} x_b(t) \xrightarrow[t\to+\infty]{} +\infty.$$

By *Equation 17*, we conclude that for all $\epsilon > 0$:

$$\frac{1}{i_b + i_d} \log\left(\frac{2i_d + i_b}{i_d(1+\epsilon)}\right) \leq \liminf_{t\to+\infty} u(t) \leq \limsup_{t\to+\infty} u(t) \leq \frac{1}{i_d + i_b} \log\left(\frac{2i_d + i_b}{i_d(1-\epsilon)}\right)$$

that concludes the proof. □

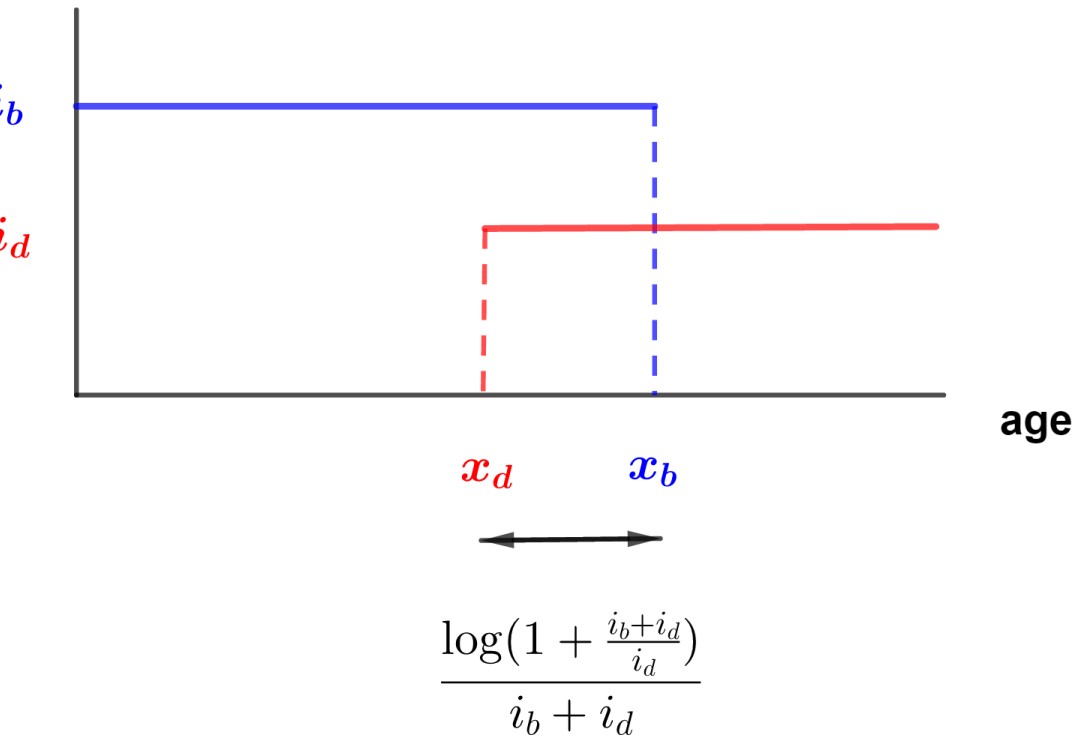

**Appendix 1—figure 3.** Optimal configurations as time tends to infinity.

## 5 On the selection of Lansing effect

In this section, we ask the question of the apparition of a pro-senescence and non-genetic mechanism similar to the Lansing effect (**Lansing, 1954**; **Lansing, 1947**). We recall that the Lansing effect is the effect through which the progeny of old parents do not live as long as those of young parents.

We will show that the Lansing effect can represent a selective advantage, as an accelerator of the evolution.

### 5.1 The bd model with Lansing effect

The bd-model with Lansing effect is defined by modifying the bd-model that we introduced in Section 1. It was introduced and studied in details in **Méléard et al., 2019** in the case where $i_b = i_d = 1$. The authors show that under the assumptions of the adaptive dynamics theory (large population, rare and small mutations), the evolution of the trait $(x_b, x_d)$ is described by the solutions a differential inclusion which reach the diagonal $\{(x_b, x_d) \in \mathbb{R}_+^2 : x_b = x_d\}$ and then stay on it. The formula given here are generalized to the case where $i_b \neq i_d$.

### The model

We assume that an individual which reproduces after age $x_d$ transmits to its descendant a shorter life-expectancy. If an individual with trait $x = (x_b, x_d)$ reproduces at age $a$, the trait of its descendant is determined by a two-phases mechanism. The first phase is non-genetic and modifies the trait $x$: if $a < x_d$ we define $\tilde{x} = x$ but if $a > x_d$, $\tilde{x} = (x_b, 0)$. The second phase corresponds to genetic mutations which modify the trait $\tilde{x}$ similarly as in Section 1. Hence, on configurations $\{(x_b, x_d) \in \mathbb{R}_+^2 : x_b < x_d\}$, the dynamics is similar as in the model described in Section 1. Let us note that the population is then composed of two subpopulations, a population with traits $\{(x_b, x_d), x_b > 0, x_d > 0\}$ and a population with traits $\{(x_b, 0), x_b > 0\}$.

### Demographic parameters

We now give for the model with Lansing effect, the analogous of the demographic parameters introduced in Section 2. We refer to **Méléard et al., 2019** for the justification. We denote by $\lambda^\ell(x)$ the Malthusian parameter describing the asymptotic growth of the population with Lansing effect. It is solution of

$$i_b \int_0^{x_b \wedge x_d} e^{-\lambda^\ell(x)a} da = 1.$$

Then it can been easily computed by Newton's method (as seen in Section 2) and the set of viability $\mathcal{V}_\ell$ is simple. It is composed of the traits $x = (x_b, x_d)$ such that

$$x_b \wedge x_d > \frac{1}{i_b}.$$

The associated stable age distribution $N_x^\ell(a) = (N_x^{\ell,1}(a), N_x^{\ell,2}(a))$ satisfies

$$N_x^{\ell,1}(a) = i_b\, e^{-i_d(a-x_d)_+ - \lambda^\ell(x)a}, \quad N_x^{\ell,2}(a) = i_b\, F(\lambda^\ell(x)) e^{-(i_d+\lambda^\ell(x))a},$$

where $F$ is some function that we don't detail here (**Méléard et al., 2019**, Proposition 3.5). The functions $N_x^{\ell,1}$ and $N_x^{\ell,2}$ describe the stable age distributions for populations with traits $(x_b, x_d)$ and $(x_b, 0)$ respectively. The generation time $G^\ell(x)$ is given by

$$G^\ell(x) = \int_0^{x_b \wedge x_d} a\, i_b\, e^{-i_d(a-x_d)_+ - \lambda^\ell(x)a} da = i_b \int_0^{x_b \wedge x_d} a e^{-\lambda^\ell(x)a} da. \tag{18}$$

We observe that the Malthusian parameter $\lambda^\ell(x)$ and the mean generation time $G^\ell(x)$ only take into account the individuals reproducing before age $x_b \wedge x_d$.

### Evolution of the trait with Lansing effect

Let us now describe the behaviour of the trait. On the subset $\{x_b < x_d\}$, the Lansing effect doesn't act. So, the dynamics is similar as the one described in the above sections. The trait dynamics is described by the differential equation

$$\frac{dx_b(t)}{dt} = \frac{\partial \lambda^\ell(x)}{\partial x_b} \frac{\lambda^\ell(x)}{2i_b\,c\int_{\mathbb{R}_+}(N_x^{\ell,1}+N_x^{\ell,2})(a)da}\sigma^2(x), \quad \frac{dx_d(t)}{dt}=0.$$

Thus, the trait $x_b$ increases while the trait $x_d$ stays constant. On the subset $\{x_b > x_d\}$, only individuals breeding before the age $x_d$ will have viable offspring. Thus, there is no selective advantage in extending the reproduction phase by increasing $x_b$, but only in increasing survival by increasing $x_d$. More precisely, on $\{x_b < x_d\}$, we have:

$$\frac{dx_d(t)}{dt} = \frac{\partial \lambda^\ell(x)}{\partial x_d} \frac{\lambda^\ell(x)}{2i_b\,c\int_{\mathbb{R}_+}(N_x^{\ell,1}+N_x^{\ell,2})(a)da}\sigma^2(x), \quad \frac{dx_b(t)}{dt}=0. \tag{19}$$

Indeed, the derivatives of the fitness are given as follows (see **Méléard et al., 2019** Proposition 4.1).

$$\forall x \in U_1, \nabla\lambda^\ell(x) = \left(\frac{i_b e^{-\lambda^\ell(x)x_b}}{G^\ell(x)}, 0\right) \; ; \; \forall x \in U_2, \nabla\lambda^\ell(x) = \left(0, \frac{i_b e^{-\lambda^\ell(x)x_d}}{G^\ell(x)}\right).$$

We observe that the trait $x_d$ increases while the trait $x_b$ stays constant. Hence, whatever the initial condition, the trait $x$ reaches in finite time the diagonal $\{x_b = x_d\}$ and then stays on it. On this diagonal the trait can evolve at different speeds (the dynamics is not unique): the global behavior of the trait is described by a differential inclusion (**Méléard et al., 2019**, Theorem 4.17).

## 5.2 Selection for Lansing effect

Let us first note that for Non Lansing and Lansing populations, as observed in the study of adaptive dynamics, the long time strategy leads to traits $x_b$ and $x_d$ going to infinity, with $x_b - x_d = \log(1 + \frac{i_b+i_d}{i_d})/(i_b+i_d)$ in the Non Lansing case and $x_b = x_d$ in the Lansing case (see **Méléard et al., 2019**, Theorem 4.17 in that case). It is then easy to deduce that in both cases, the Malthusian parameter, which has been proved to be less than $i_b$, converges to $i_b$ when $t$ tends to infinity. Therefore the evolution will give the same selective advantage to both populations, making possible the cohabitation of the two populations. In addition, we observe that the partial derivatives of the Malthusian parameters with respect to $x_b$ or $x_d$ (in both cases) are positive, meaning that the convergences are increasing. Let us consider a monotype population with trait $(x_b, x_d) \in U_2$, then by definition, we obtain that

$$\lambda^{n\ell}(x) > \lambda^\ell(x)$$

at time 0. Thus, there are periods where the Lansing fitness will increase much more than the Non Lansing one.

In order to assess the relative evolutionary success of Non Lansing/Lansing populations, we consider a population composed of two sub-monomorphic populations with traits respectively $x^\ell = (x_b^\ell, x_d^\ell)$ and $x^{n\ell} = (x_b^{n\ell}, x_d^{n\ell})$, the first one subject to the Lansing effect and the second one which is not affected by this senescence effect, both subjected to the same competitive pressure. The traits have been chosen such that the two sub-populations have the same darwinian fitness $\lambda^{n\ell}(x^{n\ell}) = \lambda^\ell(x^\ell)$. In each sub-population, the dynamics is described either in Section 1 (without Lansing effect) or in Section 5.1 (with Lansing effect). Let us first note that since $\lambda^{n\ell}(x^{n\ell}) = \lambda^\ell(x^\ell)$ and since by definition,

$$i_b \int_0^{x_d^\ell} e^{-\lambda^\ell(x^\ell)a}da = 1 = i_b \int_0^{x_d^{n\ell}} e^{-\lambda^{n\ell}(x^{n\ell})a}da + i_b \int_{x_d^{n\ell}}^{x_b^{n\ell}} e^{-i_d(a-x_d^{n\ell})}e^{-\lambda^{n\ell}(x^{n\ell})a}da,$$

we deduce immediately that

$$x_b^{n\ell} > x_d^\ell > x_d^{n\ell}.$$

We observe the isoclines of $\lambda^{n\ell}$ and $\lambda^\ell$ when they have the same values. Although they are very simple (horizontal or vertical lines) in the Lansing case, and in the region $U_1$ for the non-Lansing case, they have a more complicated form in the region $U_2$ for the non-Lansing case (**Figure 5** of the main paper).

Let us consider the points $x^{n\ell} \in U_2$ such that $\lambda^{n\ell}(x^{n\ell})$ has a fixed constant value. Using the Implicit Function Theorem, we know the existence of a real-valued smooth function $\varphi_{n\ell}$ such that for all these points, $x_d^{n\ell} = \varphi_{n\ell}(x_b^{n\ell})$. Further,

$$\varphi'_{n\ell}(x_b^{n\ell}) = -\frac{\frac{\partial \lambda^{n\ell}}{\partial x_b^{n\ell}}}{\frac{\partial \lambda^{n\ell}}{\partial x_d^{n\ell}}}(x^{n\ell}).$$

The previous computations showed that the partial derivatives of $\lambda^{n\ell}$ are positive, and then that $\varphi'_{n\ell}(x_b^{n\ell}) < 0$, yielding the function $\varphi_\ell$ to be decreasing on $U_2$. Moreover, the exact computation gives

$$\varphi'_\ell(x_b^{n\ell}) = -\frac{i_b e^{-i_d(x_b^{n\ell} - x_d^{n\ell})} e^{-\lambda(x^{n\ell})x_b^{n\ell}}}{i_b i_d e^{i_d x_d^{n\ell}} \int_{x_d^{n\ell}}^{x_b^{n\ell}} e^{-(i_d + \lambda^{n\ell}(x^{n\ell}))a} da} \geq -\frac{1}{i_d(x_b^{n\ell} - x_d^{n\ell})}.$$

The last inequality explains the almost vertical tangent observed when $x^{n\ell}$ is close to the diagonal (see *Figure 5* of the main paper).

