## [Editor Report · eLife assessment]

Through a theoretical approach, this study makes **important** contributions to our understanding of the evolutionary causes of the ageing process. Using a simple individual-based model and computational simulations, the authors provide **convincing** evidence that ageing can be a trait under natural selection, opening the door for further discussion in the context of lifespan extension research.

---

## [Referee Report · Joint public review]

Roget et al. build on their previous work developing a simple theoretical model to examine whether ageing can be under natural selection, challenging the mainstream view that ageing is merely a byproduct of other biological and evolutionary processes. The authors propose an agent-based model to evaluate the adaptive dynamics of a haploid asexual population with two independent traits: fertility timespan and mortality onset. Through computational simulations, their model demonstrates that ageing can give populations an evolutionary advantage. Notably, this observation arises from the model without invoking any explicit energy tradeoffs, commonly used to explain this relationship.

Additionally, the theoretical model developed here indicates that mortality onset is generally selected to start before the loss of fertility, irrespective of the initial values in the population. The selected relationship between the fertility timespan and mortality onset depends on the strength of fertility and mortality effects, with larger effects resulting in the loss of fertility and mortality onset being closer together. By allowing for a trans-generational effect on ageing in the model, the authors show that this can be advantageous as well, lowering the risk of collapse in the population despite an apparent fitness disadvantage in individuals. Upon closer examination, the authors reveal that this unexpected outcome is a consequence of the trans-generational effect on ageing increasing the evolvability of the population (i.e., allowing a more effective exploration of the parameter landscape), reaching the optimum state faster.

The simplicity of the proposed theoretical model represents both the major strength and weakness of this work. On one hand, with an original and rigorous methodology, the logic of their conclusions can be easily grasped and generalised, yielding surprising results. Using just a handful of parameters and relying on direct competition simulations, the model qualitatively recapitulates the negative correlation between lifespan and fertility without requiring energy tradeoffs. This alone makes this work an important milestone for the rapidly growing field of adaptive dynamics, opening many new avenues of research, both theoretically and empirically.

On the other hand, the simplicity of the model also makes its relationship with living organisms difficult to gauge, leaving open questions about how much the model represents the reality of actual evolution in a natural context. In particular, a more explicit discussion on how the specifics of the model can impact the results and their interpretation is needed. For example, the lack of mechanistic details on the trans-generational effect on ageing makes the results difficult to interpret. Even if analytical results are obtained, most of the observations appear derived from simulations as they are currently presented. Also, the choice of parameters for the simulations shown in the paper and how it relates to our biological knowledge is not fully addressed by the authors. Finally, the conclusions of evolvability are insufficiently supported, as the authors do not show if the wider genotypic variability in populations with the ageing trans-generational effect is, in fact, selected.

---

## [Author Response]

The following is the authors’ response to the original reviews.

**Public Reviews:**
Roget et al. build on their previous work developing a simple theoretical model to examine whether ageing can be under natural selection, challenging the mainstream view that ageing is merely a byproduct of other biological and evolutionary processes. The authors propose an agent-based model to evaluate the adaptive dynamics of a haploid asexual population with two independent traits: fertility timespan and mortality onset. Through computational simulations, their model demonstrates that ageing can give populations an evolutionary advantage. Notably, this observation arises from the model without invoking any explicit energy tradeoffs, commonly used to explain this relationship.

The model’s results are based on both numerical simulations and formal mathematical analysis.

Additionally, the theoretical model developed here indicates that mortality onset is generally selected to start before the loss of fertility, irrespective of the initial values in the population. The selected relationship between the fertility timespan and mortality onset depends on the strength of fertility and mortality effects, with larger effects resulting in the loss of fertility and mortality onset being closer together. By allowing for a trans-generational effect on ageing in the model, the authors show that this can be advantageous as well, lowering the risk of collapse in the population despite an apparent fitness disadvantage in individuals. Upon closer examination, the authors reveal that this unexpected outcome is a consequence of the trans-generational effect on ageing increasing the evolvability of the population (i.e., allowing a more effective exploration of the parameter landscape), reaching the optimum state faster.The simplicity of the proposed theoretical model represents both the major strength and weakness of this work. On one hand, with an original and rigorous methodology, the logic of their conclusions can be easily grasped and generalised, yielding surprising results. Using just a handful of parameters and relying on direct competition simulations, the model qualitatively recapitulates the negative correlation between lifespan and fertility without requiring energy tradeoffs. This alone makes this work an important milestone for the rapidly growing field of adaptive dynamics, opening many new avenues of research, both theoretically and empirically.

We thank the reviewers and editor for highlighting the importance of the work presented here.

On the other hand, the simplicity of the model also makes its relationship with living organisms difficult to gauge, leaving open questions about how much the model represents the reality of actual evolution in a natural context.

We presented both in results and discussion how the mathematical trade-offs between fertility and survival time give rise to (xb, xd) configuration representative of existing aging modes.

In particular, a more explicit discussion of how the specifics of the model can impact the results and their interpretation is needed. For example, the lack of mechanistic details on the trans-generational effect on ageing makes the results difficult to interpret.

We discussed the role of the transgenerational Lansing effect played to its function, there is no need for a particular mechanism beyond that function of transgenerational negative effect. We reinforce this in the discussion by adding the following sentence “Regarding the nature of the transgenerational effect, our model is agnostic and the mere transmission of any negative effect would be sufficient to exert the function.“

Even if analytical results are obtained, most of the observations appear derived from simulations as they are currently presented. Also, the choice of parameters for the simulations shown in the paper and how they relate to our biological knowledge are not fully addressed by the authors.

The long time limit of the system with and without the Lansing effect is based on analytical results later confirmed using numerical simulations. The choice of parameters is explained in the introduction as being the minimum ones for defining a living organism. As for the parameters’ values, our numerical analysis gives a solution for any ib, id, xb and xd on R+, making the choice of initial value a mere random decision.

Finally, the conclusions of evolvability are insufficiently supported, as the authors do not show if the wider genotypic variability in populations with the ageing trans-generational effect is, in fact, selected.

We do not show nor claim that evolvability per se is selected for but that the apparent advantage given by this transgenerational effect seems to be mediated by an increased genotypic/phenotypic variability conferred to the lineage that we interpreted as evolvability.

**Recommendations for the authors**
(1) The authors could use the lineage tracing results for the evolvability aspect. Specifically, within subpopulations featuring the Lansing effect, it would be valuable to explore whether individuals with parental age greater than the mortality onset (a > x_d) demonstrate higher fitness compared to individuals with a < x_d. Additionally, an examination of how this variation evolves over time could provide further insights into the dynamics of the proposed model.

We thank the reviewer for this suggestion. This is an ongoing work in the group, especially in the context of varying environmental conditions.

(2) In all simulations, I_b = I_d = 1, resulting in total fertility (x_b * I_b) equating to x_b, while x_d is proportional to life expectancy. Considering an exploration of the implications of this parameter setting, the authors could frame x_d as a 'lifespan cost', potentially allowing for the model to be conceptualised in terms of energetic tradeoffs. This might offer additional perspectives on the dynamics of the model and its alignment with biological principles.

We discuss how the apparent trade-offs given by the model depending on ib and id values can be related to the interpretation of such trade-offs that has been accepted for most of the past century. Our claim here in the discussion is that one does not need such energetic trade-off for the fertility/longevity trade-offs to appear. Such energetic trade-off is not a “biological principle” but merely an accepted interpretation of a fertility/longevity trade-off that is not even a general mechanism.

(3) Considering the necessity of variation in x_d for the observed patterns, an exploration could be undertaken by the authors to examine a model where x_d is simply variable without inheritance. This could involve centring x_d at some value d with some variance σ_d for all individuals. In such a scenario, it may be observed whether the same convergence of x_b - x_d occurs without requiring x_d to be selected. Furthermore, similar consequences of the Lansing effect could potentially be identified.

This was done early on during our work and did not show any major changes in the model’s behaviour beyond the time of convergence. We did not include it to the final manuscript because of the low added value to an already long and complex manuscript.

(4) While it may not be necessary to alter the model itself, it is suggested that the authors consider acknowledging the potential consequences of certain modelling decisions that might be perceived as biologically unrealistic. Notable examples include assumptions such as fertility from birth and zero mortality prior to x_d. These assumptions, such as infertility from birth, could be viewed as distinctive features, and it might be worth mentioning that parental care of offspring could have co-evolved with such features. This is particularly relevant considering the energy tradeoff hypothesis that has been postulated.

Although inspired from results obtained in *Drosophila*, mice, nematodes and zebrafish, the model is so far haploid and asexual, thus involving individuals likely more similar to unicellulars. In these conditions, infertility from birth did not seem relevant to us. However, the model and codes are accessible online and we hope that others will use it to address such questions. It is interesting though to notice that ageing appears here without such constraint.

Additionally, the consideration that all organisms face a non-trivial mortality rate at every age, not solely from physiological causes, reflects the reality within which selection operates.

We thought this was the best way to reflect, an environment with a limited carrying capacity. A more complex model is under construction to take into account the fact that older individuals might be more sensitive to it than younger ones.

(5) While acknowledging the technical rigour applied by the authors, it is suggested that further attention be given to conducting a comprehensive 'reality check' associated with the chosen parameters, particularly regarding the biological relevance of the results. For instance, the authors argue that offspring of old organisms do not, on average, live similarly to their parents. However, it is noted that studies in the haploid asexual organism yeast, akin to what the authors model (albeit not necessarily yeast), revealed that the average lifespan of yeast progeny born from young or old mothers is very similar.

We do not claim that progeny of old parents live less long than that of younger parents on average, we say that it happens in the progeny of physiologically old parents, representing at most 10% of the population in our numerical simulations.

The authors cite experimental evolution in *Drosophila* progeny conceived later in the life of the parent, indicating that the onset of mortality in these progeny occurs late, sometimes even after the end of the fertility period (Burke et al., 2016; Rose et al., 2002). While the authors report their own previous studies with divergent results, independent experiments have suggested an increase of x_d following an artificial increase of x_b (Luckinbill and Clare, 1985; Sgro et al., 2000). A more in-depth consideration of these contrasting observations and their potential implications for the current model could enhance the overall robustness of the study.

The increase of x_d following an artificial increase of x_b is predicted by our model as discussed. The divergence of observations between studies is alas hard to assess.

(6) To enhance readability and maintain consistency, it is suggested that the authors homogenise the description of key parameters, specifically x_b and x_d, throughout the text. This could contribute to improved clarity and rigour. One recommendation is to refer to x_b consistently as the 'fertility span' and x_d as the 'mortality onset' for the sake of uniformity in terminology.

We have modified the text accordingly.

(7) At various points in the text, the assertion is made that observations have indicated a tradeoff between fertility and longevity. It is recommended that the authors provide references or data to substantiate this claim. This addition would contribute to the empirical grounding of the mentioned tradeoff and strengthen the overall support for the assertions made in the study.

We added the following references to the discussion Lemaitre et al., 2015, Kirkwood, 2005 and Rodrigues and Flatt 2016.

(8) The statement claiming that the model is 'able to describe all types of ageing observed in the wild' should be moderated. As the authors themselves acknowledge, the model is referred to as a 'toy model,' and it is made clear that it cannot capture, nor is intended to capture, the entire diversity observed in life. Adjusting this statement to reflect the limited scope and purpose of the model would enhance precision and accuracy in the presentation of its capabilities.

Although a toy model, its possible configurations encompass all the possible configurations described so far across the diversity of ageing throughout the tree of life from negligible senescence with no loss of fertility (x_b and x_d >> 0) to menopause-like configurations (x_b >> x_d) through fast mortality increase post reproduction (x_b = x_d). Replacing our current square functions would allow age-dependant decrease or increase of fertility and/or risks of mortality onsets.

(9) To bolster the biological relevance of the study, it is strongly recommended that the authors cross-check the results of their simulations with previously published experimental findings. This approach would serve to strengthen the alignment between the model outcomes and observed biological phenomena. Additionally, placing greater emphasis on the biological relevance aspects throughout the text would contribute to a more robust and comprehensive exploration of the study's implications.

In the present manuscript we have tried to cite a certain number of results from artificial selection experiments on life history traits in order to strengthen the interpretations of our model’s behaviour. There are numerous other studies, going in the same direction or not, but we do not think that it would be relevant to add an exhaustive list of them. Nevertheless, we added Stearns et al., 2000 that adds extrinsic high mortality to the evolution of life history traits.

(1) For enhanced clarity, it is suggested that the x-axis in Figure 1 be labelled as 'age.' Considering this adjustment could contribute to clearer visual communication of the data.

We agree with the reviewer and modified the figure accordingly.

(!!) The addition of graphical legends is recommended for Figures 3-5, as well as the supplementary figures. Including these legends would provide essential context and improve the interpretability of the figures for readers.

We agree with the reviewer and modified the figure accordingly.

(12) For improved distinction of the ranges indicated by quantiles in Figure 3, it is suggested that the authors consider enhancing visual clarity. One approach could involve making the middle quantile thicker or using a different line type. Additionally, it is recommended to explore the calculation of the highest density 90% intervals rather than the 1-9 deciles. This adjustment could contribute to a clearer representation of the data distribution in the figure.

We named the different deciles directly on the figure to improve readability.

(13) It is observed that the mathematical proofs in Annex 1 are not displaying properly in the PDF. Additionally, there seem to be missing and broken references for the Annex. This issue may be related to LaTeX formatting. The authors could consider revisiting the formatting of Annex 1 to ensure the correct display of mathematical proofs and address the referencing concerns, possibly by checking and rectifying any LaTeX-related issues.

The latex file of the supplementary was not correctly compiled. It is now corrected.

(14) There is inconsistency in the text regarding the reference to the Annex, with both 'Annex' and 'Annexe' being used interchangeably. To maintain uniformity, it is suggested that the authors consistently use either 'Annex' or 'Annexe' throughout the text. This adjustment would contribute to a more polished presentation of the supplementary material.

We corrected them accordingly.

(15)There appears to be a typographical error in the name of Supplementary Figure 3.

We corrected it accordingly.